# Characterizing the structural complexity of the Earth's forests with spaceborne lidar

Tiago de Conto [1], John Armston[1] & Ralph Dubayah [1] ✉

Forest structural complexity is a key element of ecosystem functioning, impacting light environments, nutrient cycling, biodiversity, and habitat quality. Addressing the need for a comprehensive global assessment of actual forest structural complexity, we derive a near-global map of 3D canopy complexity using data from the GEDI spaceborne lidar mission. These data show that tropical forests harbor most of the high complexity observations, while less than 20% of temperate forests reached median levels of tropical complexity. Structural complexity in tropical forests is more strongly related to canopy attributes from lower and middle waveform layers, whereas in temperate forests upper and middle layers are more influential. Globally, forests exhibit robust scaling relationships between complexity and canopy height, but these vary geographically and by biome. Our results offer insights into the spatial distribution of forest structural complexity and emphasize the importance of considering biome-specific and fine-scale variations for ecological research and management applications. The GEDI Waveform Structural Complexity Index data product, derived from our analyses, provides researchers and conservationists with a single, easily interpretable metric by combining various aspects of canopy structure.

One of the most important variables emerging in ecological research of forested ecosystems is structural complexity. While definitions of structural complexity vary, they largely converge around its characterization as a function of the heterogeneity of canopy structure in 3D space. This heterogeneity is determined by tree diameter, height, biomass, leaf angle, deadwood, and vertical layering, along with their relative abundance and spatial arrangement, among others[1–5]. Recent work has explored the development of theoretical frameworks that address both the ecological role of complexity in ecosystem function[6] and composition[7,8], as well as the environmental and anthropogenic drivers that shape complexity[9]. These studies have been aided by advances in remote sensing methods, particularly lidar, that can detail the 3D structure of forest canopy elements in ways that were hitherto unobtainable.

Ecosystem productivity, water use strategies, and carbon use efficiency are largely explained by traits of vegetation structure directly linked to structural complexity, including aboveground biomass, tree height, and leaf area index[6]. Forest structural complexity may also be an effective surrogate to measure and monitor biodiversity and restoration effectiveness, as forest structure regulates microclimate conditions and provides niche space for organisms, thereby enabling functional redundancies that increase resistance and resilience to both natural and anthropogenic disturbances[10].

Structural complexity is underpinned by detailed measures of multiple forest structural traits. Some of these may be readily quantified at the field plot level, such as stem diameters and tree heights, but are difficult to map across landscapes given the limited number and distribution of forest inventory plots globally. Other important elements of structure, such as vertical stratification and layering, tree architecture, and the horizontal and vertical distribution of canopy gaps are more difficult to measure in the field, and as the level of detail in the field inventory increases, the smaller the extent the inventory can cover[4]. Consequently, characterization of complex forest structural traits over large areas often must rely on remote sensing data and while some traits, such as height, may be measured directly others, such as biomass, must be inferred from relationships among field

---

[1]University of Maryland, 2181 Samuel J. LeFrak Hall, 7251 Preinkert Drive, College Park, MD, USA. ✉e-mail: dubayah@umd.edu

measurements and the remotely sensed observation. While passive optical and radar remote sensing have expanded the reach of forest structure assessments, these have been shown to not be sufficiently sensitive to important variations in 3D canopy structure[10,11]. In contrast, lidar remote sensing has emerged as the most effective means for forest structure assessments, and due to its direct 3D mapping capabilities and canopy penetration, can observe not only horizontal structural variability but also the vertical variability of the canopy[12,13].

There are several different types of lidar used for forest canopy characterization. Terrestrial Laser Scanning (TLS) provides remarkable 3D data at very high spatial resolutions, allowing digital reconstruction of structural elements and gaps at fine scales (on the order of centimeters)[14,15], but is time-consuming, restricted to small sites, and suffers from signal saturation in tall canopies. Airborne Laser Scanning (ALS) and drone-based lidar provide structural measurements of forests and are used to map forest heights, canopy cover, plant area index (PAI), foliage height diversity (FHD), and other variables over larger sites[13]. However, the limited spatial extent, lack of global availability, and fragmentation among ALS and TLS datasets, especially in the tropics, hinder the characterization of complexity and its variability within and among biomes. Thus, the challenge of mapping structural complexity consistently at global scales remains.

Progress in advancing our knowledge of the importance of structural complexity on the functioning and resilience of ecosystems is underpinned by the ability to measure complexity far more widely than currently exists. This is because we do not yet fully understand how complexity develops, how horizontal and vertical complexity are linked, nor how the myriad of environmental and biological factors interact to control its spatial distribution and evolution. Ehrbrecht et al.[16] explored some of these factors by modeling a stand structural complexity index (SSCI_{pot}) that related climatic and edaphic variables and TLS point clouds at 1 km resolution. The resulting map depicts potential complexity, that is what the complexity could be in a specific location, and not the actual structural complexity that currently exists there. The impacts of disturbance and subsequent recovery could not be assessed, even though these are important determinants of the actual complexity[16,17]. Furthermore, their models relied on 294 TLS point clouds distributed across 20 sites in different continents, producing a global map extrapolated from a small pool of training samples. Although their models underwent cross-validation, the limited number of samples restricted the extent of validation, which could contribute to the overgeneralization of complexity as a function of climate variables.

One means of obtaining the required data on forest structure as it currently manifests on the land surface is using spaceborne lidar. The Global Ecosystem Dynamics Investigation (GEDI) mission was optimized to monitor ecosystem structure of the Earth's tropical and temperate forests, collecting data along the orbital track of the International Space Station (ISS), between ±51.6° latitude[18]. GEDI waveforms record the vertical distribution of leaves and branches from the top of the canopy to the ground over 25 m diameter footprints. In operation from April 2019 to March 2023, GEDI has acquired billions of observations over the land surface. The GEDI observable is a return waveform that provides a direct measure of vertical variability through relative height (RH) metrics[19]. GEDI waveforms are also used to derive a variety of other structural properties, including canopy height, canopy cover, plant area index, aboveground biomass, and others[18] within each footprint. Given these properties, GEDI data are well suited for mapping actual structural complexity on a global scale.

Lidar remote sensing of canopies can produce vast amounts of data leading to many different descriptors of canopy structure. This has led to the creation of indices of complexity that provide compact yet meaningful summaries of structural variability that may be readily mapped and interpreted by ecologists[2,20–23]. Additionally, the use of a single index facilitates the incorporation of structural complexity into ecological modeling and applications[4]. While GEDI produces less data over its 25 m footprint than a TLS or ALS survey, its waveforms are nonetheless described by 100 RH metrics at each footprint. Hence, our motivation is to provide an index of complexity from GEDI waveforms. Furthermore, GEDI only records vertical variations in canopy structure, as given by variations in the amplitude of its returned waveform at any given height. There is thus considerable interest to assess the degree to which 3D complexity may be inferred from vertical variations in the waveform alone, under the hypothesis that vertical and horizontal complexity must be related.

Here we develop a footprint-scale Waveform Structural Complexity Index (WSCI). This index is established by modeling the empirical relationships between GEDI RH metrics and an existing metric of 3D structural complexity ($CE_{XYZ}$)[20] derived from ALS point clouds. $CE_{XYZ}$ is an entropy-based measure that captures both horizontal and vertical complexity. Horizontal complexity refers to the spatial distribution of canopy structures within a footprint, while vertical complexity describes the distribution of vegetation layers from the ground to the canopy top. Together, these components provide a comprehensive characterization of the 3D structural complexity of the forest canopy. We create global models using over 800,000 measured values of $CE_{XYZ}$ from ALS and collocated GEDI data to predict WSCI for four different plant functional types (PFTs) at the scale of GEDI footprints. We then apply these models to approximately 2 billion GEDI shots and use the resulting WSCI values to create a data set of existing structural complexity for the Earth's temperate and tropical forests. We analyze which RH metrics are most important for modeled WSCI, identifying whether these originate from the lower, middle, or upper layers of the waveform, towards understanding how different elements of the canopy impact complexity and how these vary by biome[24]. Our analysis reveals that highly complex canopies are concentrated in tropical broadleaf forests, and this complexity appears to be more strongly driven by variability in metrics linked to the lower and middle waveform layers than in other forest types. Additionally, we find strong evidence of scaling relationships between structural complexity and canopy height globally, but these vary geographically and by biome. Tropical forests tend towards having smaller scaling exponents but their complexity in early seral growth stages surpasses that of other biomes; e.g. short tropical forests show higher complexity relative to other forests of similar heights. These results emphasize the value of widespread and consistent estimates of forest structural complexity and their variability while opening pathways for investigations on how forest structural complexity develops as a result of gradients in environmental drivers, seral stage, and disturbance conditions. Our results confirm the complexity of tropical forests, and thus their conservation significance, but also highlight the importance of considering the protection of what GEDI data reveal to be rare yet highly complex forests in temperate biomes.

## Results

### Model performance and feature importance

We created regression models to predict WSCI using extreme gradient boosted trees (XGBoost)[25] for four plant functional type (PFT) models: Evergreen Broadleaf Trees (EBT), Deciduous Broadleaf Trees (DBT), Evergreen Needleleaf Trees (ENT), and Grasslands, Shrublands and Woodlands (GSW). The models were trained using GEDI RH percentiles as independent variables to predict the 3D structural complexity index ($CE_{XYZ}$)[20] derived from ALS point clouds collocated with GEDI footprints. This training approach employed a thorough spatial cross-validation method to strive for broad geographical applicability. Our models explained 68% of the variability of complexity in the training data. As part of our analyses, we also created separate models to predict horizontal and vertical complexity at the PFT level. This approach helped us determine if horizontal canopy structure, which GEDI waveforms do not measure, can be inferred based on its

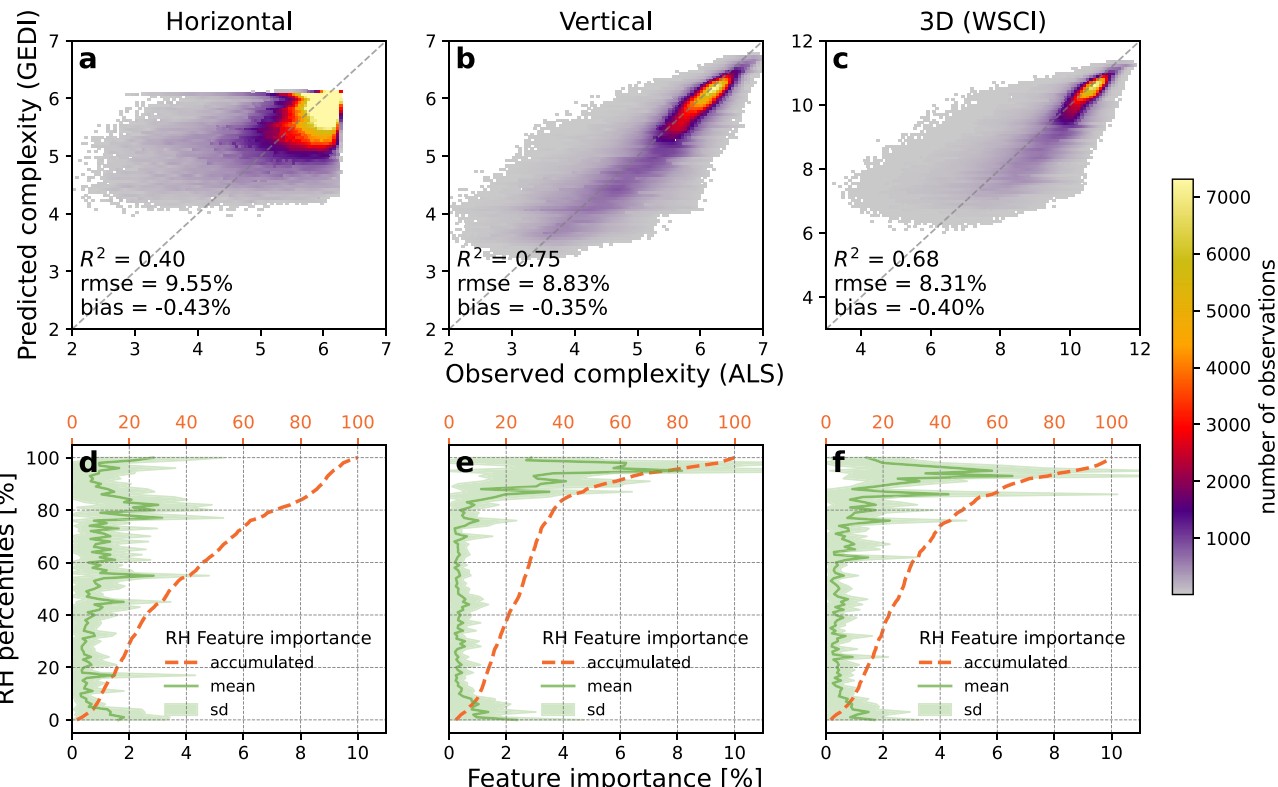

**Fig. 1 | Performance and feature importance profiles of models using GEDI for predicting horizontal, vertical, and 3D structural complexity.** Measures from Airborne Laser Scanning (ALS) data were modeled using relative height (RH) percentiles from collocated GEDI footprints ($n = 816,275$). The models (**a**, **d**) infer horizontal complexity using information distributed throughout the entire RH profile, i.e. drawing information from both the lower and upper canopy, **b**, **e** while vertical complexity uses larger RH percentiles nearer to the top of the canopy. Total 3D complexity measured by the Waveform Structural Complexity Index (WSCI)

(**c**, **f**) combines RH features from both horizontal and vertical complexity models. The color gradients in (**a**, **b**, **c**) represent the number of samples and the dashed lines represent a perfect relationship between observed and predicted structural complexity. The solid lines in (**d**, **e**, **f**) represent mean feature importance at a given RH percentile, shaded areas represent their standard deviation, and the dashed lines represent the mean accumulated feature importance from 0% to 100% relative height.

relationship with vertical structure. We found that the vertical complexity was well-predicted ($R^2 = 0.75$, RMSE = 8.8%), with lower performance for horizontal complexity ($R^2 = 0.40$, RMSE = 9.5%) (Fig. 1). GEDI does not measure variability of horizontal structures within individual footprints, but instead integrates the return signal from all leaves and branches at any particular height into one value, which is represented by the cumulative RH metric at that height. Our results suggest that horizontal and vertical canopy structures are linked at the footprint scale, confirmed by the strong relationship between horizontal and vertical complexity observed in our ALS samples (Supplementary Fig. 1), which enables some variation in horizontal complexity to be inferred from vertical structure co-variates. Note that maximum horizontal complexity is limited by footprint size (25 m), hence the sharp cut-off in values at 6.3 shown in Fig. 1. In contrast, vertical complexity is limited by maximum tree height, which varies among footprints, and so does not show such behavior.

The feature importance of individual model predictions was examined using SHAP values (Shapley Additive Explanations)[26] and showed that vertical complexity is largely predicted by metrics related to upper strata features, with 46% of the feature importance accounted for by RH percentiles in the top 10% of the canopy (that is above 90% of total canopy height) (Fig. 1e). In contrast, horizontal complexity was estimated by a larger pool of canopy features at intermediate height layers, with only 12% of the feature importance accumulated in the top 10% RH percentiles (Fig. 1d). The WSCI models (which integrate both vertical and horizontal structure) showed intermediate patterns of feature importance between the horizontal and vertical complexity

models, mixing features from both canopy top and intermediate height layers among the strongest predictors, with the top 10% RH percentiles concentrating 35% of the feature importance (Fig. 1f). These results further imply a functional dependency between vertical and horizontal structural attributes in the forest's overall complexity development. As canopy height increases, vertical layering also develops, such as through the recruitment of new individuals into the understory, diversifying the tree composition, increasing tree density[3,27], and consequently affecting the forest's horizontal and vertical complexity.

Model performance was consistent among the forested PFTs explaining $65 \pm 6\%$ (RMSE), $61 \pm 8\%$, and $58 \pm 8\%$ of $CE_{XYZ}$ variability for EBT, EDT, and ENT, respectively (Fig. 2). The model fitted for GSW showed lower performance capturing $36 \pm 13\%$ of variability. Exploring the model residuals geographically, we observed low errors and little bias in our training samples over regions where forested PFTs are prevalent (Supplementary Fig. 2). Higher errors were consistently observed in samples at regions of sparse tree cover where GSW is prevalent, with overestimation noted in the midwestern United States, northern Spain, and underestimation in central Australia and South Africa (Supplementary Fig. 2). The models for ENT and GSW predicted WSCI mostly from features near the canopy top, with 68% and 48% of the importance concentrated in the top 10% RH percentiles, respectively. The broadleaf PFT models, conversely, place higher importance on features farther down in the waveform for predicting structural complexity, with only 20% of the feature importance coming from the top 10% RH percentiles.

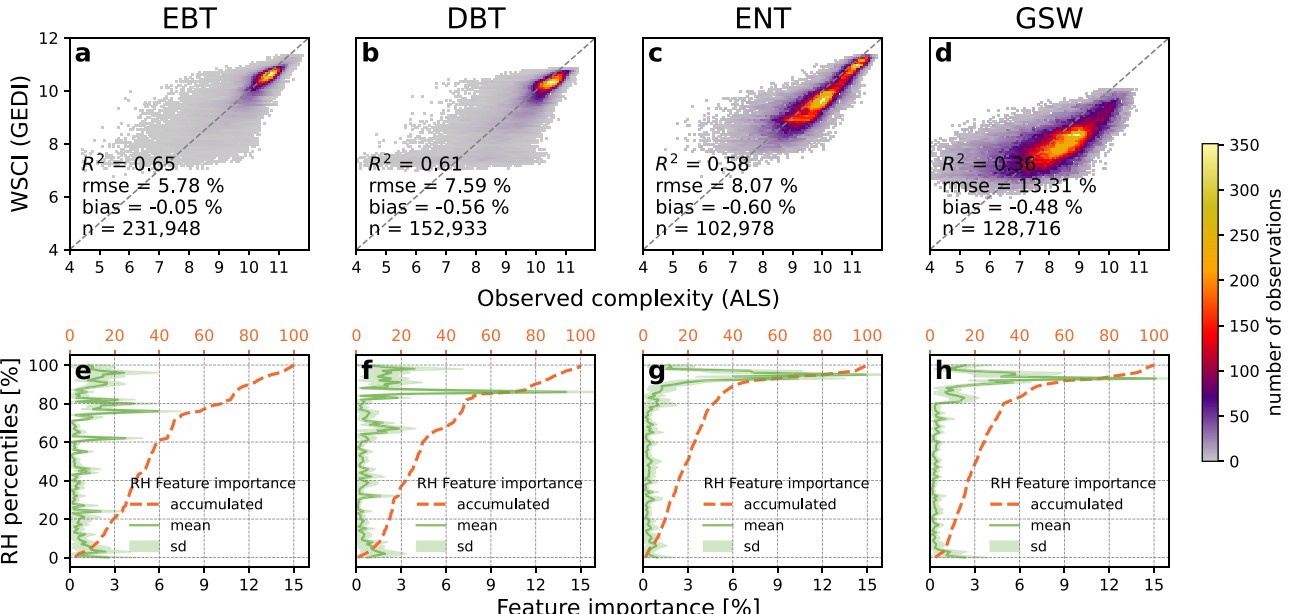

**Fig. 2 | Performance and feature importance profiles of models using GEDI for predicting the Waveform Structural Complexity Index (WSCI) of different plant functional types (PFTs).** Models trained on (**a**, **e**) evergreen broadleaf trees (EBT) (**b**, **f**) deciduous broadleaf trees (DBT) and (**c**, **g**) evergreen needleleaf trees (ENT) showed consistently better performance compared to the (**d**, **h**) grasslands, woodlands and shrublands (GSW) model. ENT and GSW models predict complexity largely from canopy top features, using relative height (RH) percentiles close to 100%, while the EBT and DBT models rely more on RH metrics lower in the waveform. The color gradients in (**a**, **b**, **c**, **d**) represent the number of samples and the dashed lines represent a perfect relationship between observed and predicted structural complexity. The solid lines in (**e**, **f**, **g**, **h**) represent mean feature importance at a given RH percentile, shaded areas represent their standard deviation, and the dashed lines represent the mean accumulated feature importance from 0% to 100% relative height.

## Global Patterns of structural complexity and model uncertainty

We applied the WSCI models to the 2 billion high-quality GEDI shots and averaged these to 1 km resolution. Doing so, we observe a latitudinal pattern of high structural complexity over the tropics, decreasing towards higher latitudes (Fig. 3a). Hotspots of high complexity are found at tropical forests, as in the Amazon, Gabon, and between Borneo and Papua. The median structural complexity in tropical forests is higher than 82% of the estimates for temperate forests, and 96% of those in other forest types (see Supplementary Fig. 3 for the definition of these forest types). Spatial gradients of complexity decrease from dense tropical forests towards sparse tree cover areas between the Amazon and the Brazilian Cerrado, and between the Congo Basin's rainforests and African savannahs. At smaller extents, hotspots of structural complexity emerge in other tropical, subtropical, and temperate forest biomes, such as the Atlantic coast rainforest, in Brazil, the region between Bhutan and Northern Myanmar, in the Himalayan Forests, the Sierra Nevada in the northwestern United States, the region between Colombia and Costa Rica, in Central America and the southeastern Australian coast. Model uncertainty was quantified using conformal predictors to calculate prediction intervals at a 95% confidence level for all WSCI estimates (detailed in Supplementary Table 1). Prediction intervals were generally inversely related to WSCI estimates (Fig. 3b), reflecting the heteroscedasticity observed in the WSCI model residuals, where the variance of the errors increase as observed complexity decreases. The larger uncertainty of the GSW model relative to the models trained on forest PFTs is also evident (Fig. 2).

The prediction of structural complexity based on vertical canopy strata also revealed distinct geographical patterns. We analyzed SHAP values derived from all WSCI estimates, dividing the accumulated feature importance into three layers for global visualization: lower (<= RH 33), middle (RH > 33 and RH <= 66), and upper (RH > 66) returns. Averaging the relative feature importance of these layers at 10 km scale (Fig. 4) reinforces that global structural complexity is largely determined by returns from the upper layer. However, in tropical forests, which are dominated by evergreen broadleaf trees, complexity relies more heavily on a combination of features from lower and middle layers. In contrast to these forests, complexity in temperate forests depends more strongly on metrics from the upper layer, but with the increased influence of middle layers in regions dominated by broadleaf trees. Regions with sparse tree cover, where GSW is prevalent, exhibit structural complexity arising from a blend of features from throughout the vertical profile. Examining feature importance profiles as a function of plant functional types sampled from forests in different geographical areas, for example, comparing EBT in the Amazon vs. Southeast Asia, largely illustrates the consistency of these findings, while also allowing for some variation in feature importance by region (Supplementary Fig. 4).

## Characterizing complexity relative to other structural elements

We expect the WSCI to be strongly associated with other measures of structure derived from waveforms, especially height and FHD; the former reflecting taller canopies' potential to encompass more 3D space in which elements such as canopy layering may develop, and the latter because it is an entropy-based measure (but in one dimension only). It is of interest to understand deviations from these relationships, e.g. having short canopies and high complexity and vice versa. Furthermore, other measures of the canopy structure, in particular, cover and plant area index (PAI) also may be indicative of complexity in canopies. GEDI derives all of these from the same RH metrics, and a single index of complexity, in our case the WSCI, likely incorporates variation from all of these.

We performed a Principal Components Analysis (PCA) to understand the relationships among WSCI and these other metrics using a 1% random sample ($n = 19,041,737$) of GEDI footprints (Fig. 5a). The first component (PC1) explains the majority of the variance in the data set (89%) and is aligned with increasing forest density, measured by cover and PAI, noting that these are highly related (PAI is a monotonic

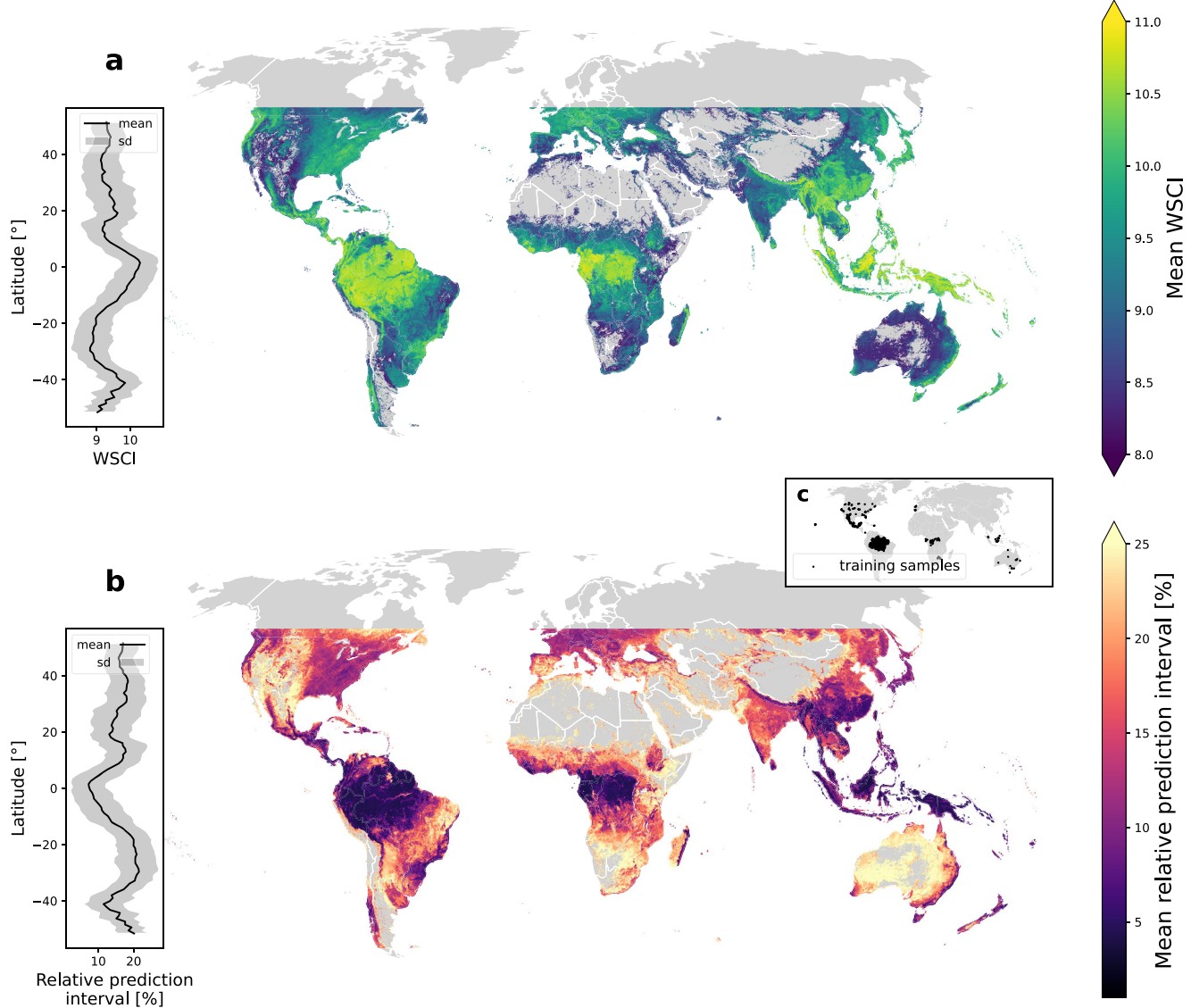

**Fig. 3 | Global patterns of Waveform Structural Complexity Index (WSCI) and model uncertainty.** Mean (**a**) WSCI, with color gradient ranging from low to high mean structural complexity, and (**b**) mean relative prediction interval size at 95% confidence were calculated at 1 km spatial resolution and resampled to 10 km for better visualization, with color gradient ranging from low to high relative model uncertainty. Latitudinal gradients (left bar on each plot) display mean (solid line) and standard deviation (shaded envelope) of WSCI and its relative prediction interval in 1-degree intervals and (**c**) shows the locations of training samples used for fitting the WSCI models.

transformation of cover, for example). The two clusters along this axis correspond to short, open woodlands at one end (low negative PC1 scores), and denser forests at the other (high positive PC1 scores) (Supplementary Fig 5a). This PC geographically captures broad-scale differences among regions, such as between the Amazon basin and the Brazilian Cerrado to the south but does not capture variability within these regions as their PC1 scores are relatively uniform. In contrast, PC2 is loaded heavily on measures explicitly linked to vertical stratification and canopy height: FHD, RH98, and WSCI and captures variations in structure within these broad regions (Supplementary Fig. 5b). As expected WSCI is aligned closely with FHD and RH98 in this PC space. WSCI is further strongly ($R^2 = 0.71$) and linearly related to FHD (Fig. 5b), and non-linearly related to RH98 (Fig. 5e). However, WSCI also incorporates some element of horizontal complexity (recalling that our models explained about 40% of the variability in horizontal complexity from ALS), comprehensively integrating structural information from GEDI waveforms that cannot be captured by any other single metric, explaining between 60% – 80% of the variability of each

variable assessed in the PCA (Supplementary Fig. 6). Note that for any given height range, there may be a large range of WSCI values (Fig. 5d) and this range compresses as canopies get taller.

The question arises as to whether WSCI, as an index, captures any variation in complexity that is not already captured with existing standard GEDI metrics (RH98, AGBD, FHD, cover and PAI). The advantage of using a single index is well established as it facilitates the incorporation of structural complexity into ecological modeling and applications[4]. In addition, because WSCI uses the entirety of the waveform and infers horizontal variability, it is of interest to establish if it derives aspects of structure not captured by these other metrics (Supplementary Fig. 6). To assess this, we used the 3 first PCA components (which accounted for 98.8% of the total variance in the dataset) to perform a principal components regression (PCR) to predict WSCI. The PCR explained 96% of WSCI's variance with the unexplained 4% likely due to nonlinear interactions among the variables not accounted for in principal components space. We then removed WSCI from our principal components analysis and found that the resulting

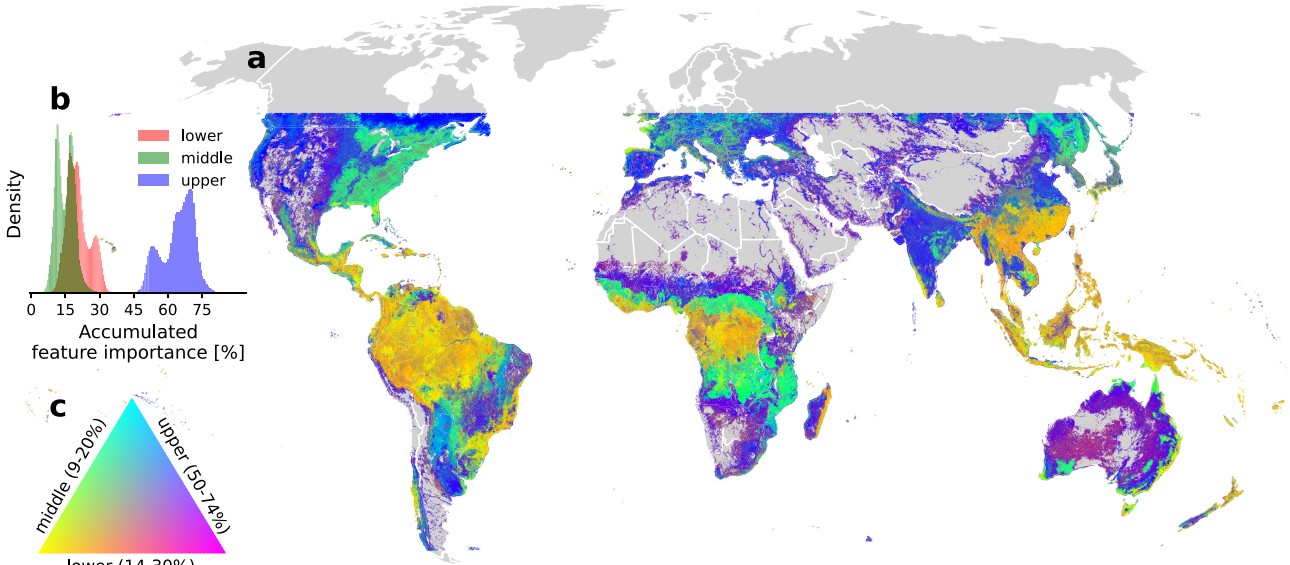

**Fig. 4 | Geographical patterns of feature importance used for predicting the Waveform Structural Complexity Index (WSCI).** (**a**) The red, green, blue (RGB) color composite maps the relative contribution of 3 canopy layers defined from relative height (RH) metrics for predicting structural complexity at 10 km resolution, **b** the band histograms show the range and distribution of accumulated feature importance of each canopy layer on a global scale, and (**c**) the color triangle displays the geographical gradients of accumulated feature importance (within their respective ranges) from all 3 canopy layers combined as RGB bands, with red

representing the lower canopy layer, green the middle layer, and blue the upper layer. Complexity in tropical forests is highly weighted on a mix of features from middle and lower layers (yellow), while complexity in temperate forests can be attributed either to the middle (green) or upper (blue) layers, depending on the geographical region. In areas of sparse tree cover, where grasslands, shrublands, and woodlands are prevalent, structural complexity is expressed by a mixture of attributes from the lower and upper layers (purple).

first 3 principal components explained more than 99.5% of the variation in the data set, with a subsequent PCR of these 3 components explaining 85% of the variance in WSCI. Thus, based on these PCR analyses, the WSCI captures 11% of the variation not included in these other metrics. That said, it is possible that new models trained with these metrics and that account for nonlinear relationships could predict $CE_{xyz}$ nearly as well. However, some of these metrics, such as AGBD[28] and PAI[29] rely on empirical calibrations and/or assumptions about canopy and ground reflectivity in their derivations, respectively. Using RH metrics directly avoids such issues and intrinsically provides a means to assess the contribution of elements of canopy structure as captured by the direct, cumulative energy returns at various heights to complexity.

### Scaling of complexity with height

Recent work has explored power law scaling relationships between various measures of structural complexity derived from ALS to canopy height across the United States[4]. While our results show that for a particular canopy height range, the range of WSCI values may be large, the trend towards higher complexity with height is evident (Fig. 5e). GEDI data enable us to derive scaling relationships globally and to assess how these may vary geographically and by PFT. We used linear regression models developed within 10 km pixels distributed over the GEDI domain to measure the scaling of WSCI as a function of canopy height within those pixels. Our analyses showed that the relationship between WSCI and canopy height may be described by a power law in most regions, with WSCI scaling linearly with log of RH98 (Fig. 6a). Deviations from linear relationships may be found within and among forest biomes, mostly in landscapes of sparse tree cover and transitional zones (Fig. 6c). We found average scaling exponents of 1.04, 1.29 and 1.33 for Tropical, Temperate, and Other biomes, respectively (Supplementary Fig. 7).

While the rate of increase in complexity per unit of canopy height is lower in tropical forests relative to other forest biomes, the initial state of structural complexity, inferred from the intercept in the

scaling relationship, is higher in tropical forests, particularly in the Amazon (Fig. 6b). This implies that short forests are consistently more complex in tropical biomes, but differences among biomes decrease with increasing height, converging near a height of 45 meters (Supplementary Fig. 7c). This phenomenon can be further observed by weakened latitudinal gradients and a more homogeneous global distribution of structural complexity when mapping complexity of only tall forests (Supplementary Fig. 8), making it harder to identify hotspots. In summary, tall forests tend to be highly complex everywhere, and what boosts structural complexity in tropical forests, relative to other biomes, is the complexity of their short forests.

## Discussion

The observed geographical patterns of structural complexity, height scaling, and associated differences in which parts of the canopy are used to predict it across biomes provide a near-global view of complexity as measured from space at relatively fine spatial scales (25 m GEDI footprints). Our models relied on different configurations of relative height metrics to explain the complexity associated with different canopy vertical layers, which had not been quantified over large scales. While these RH metrics are correlated amongst themselves[30], these linkages decrease as the vertical distance among them increases, and thus our three, broad strata of upper, middle, and lower waveform returns are likely discriminating on actual differences in structure and not artifacts of the modeling process or the data used to derive it.

Structural complexity is inherently linked to 3D space[1,2,4], hence our choice of a metric that separates complexity into vertical and horizontal components[20]. Metrics such as FHD ignore the horizontal component and are highly dependent on the number of height bins, which is determined by the top height of the canopy. GEDI waveforms only provide a direct measurement of the vertical component, leaving the horizontal component to be inferred indirectly through our model-based approach. This approach is analogous to the modeling of above ground biomass (AGB), which relies on in situ plot measurements to train prediction models based on measurements (e.g. height) that are

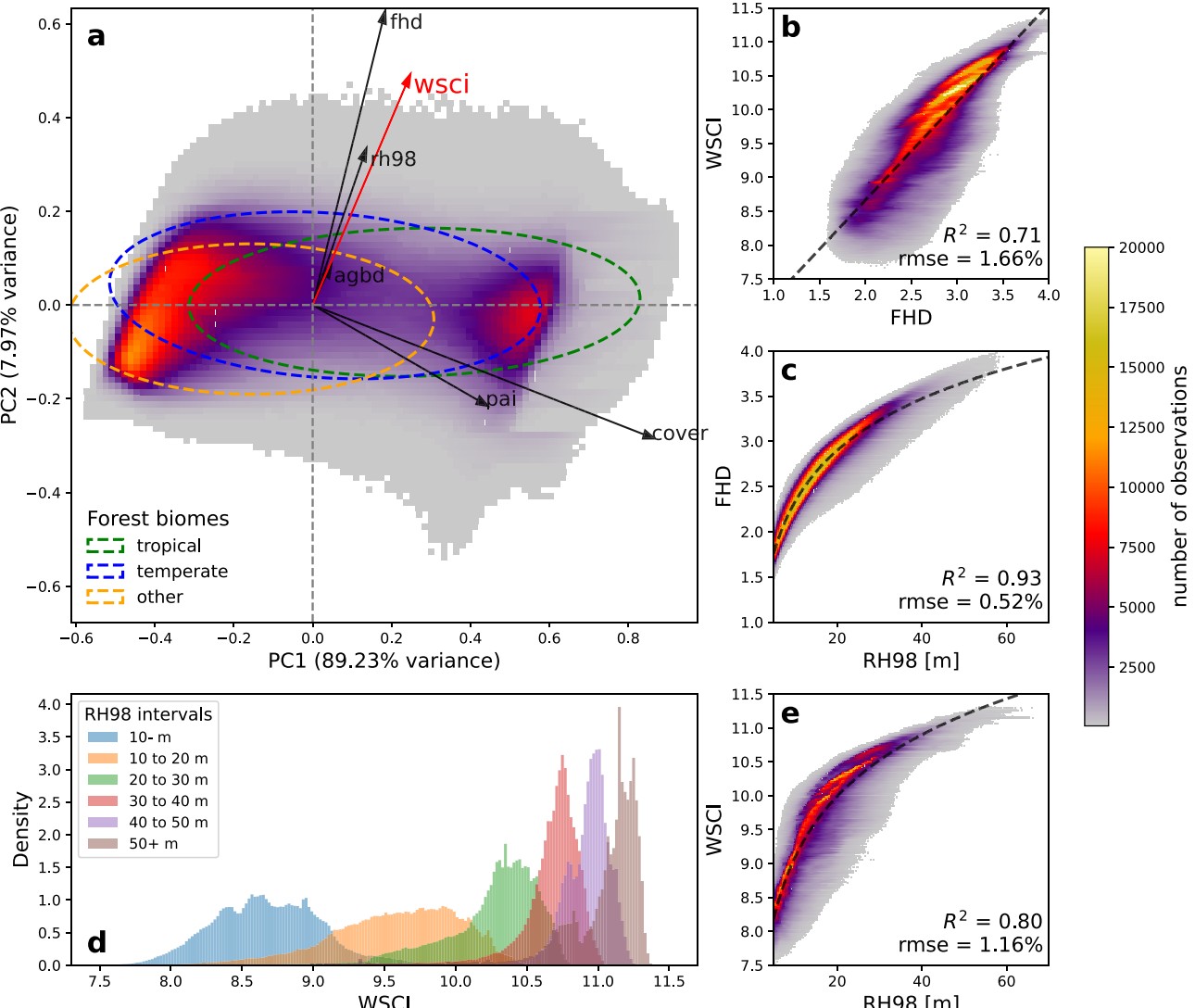

**Fig. 5 | Global relationships of GEDI structural metrics. a** Principal Components Analysis (PCA) of GEDI structure metrics overlaid by 90% confidence ellipses for Tropical, Temperate and Other Forest biomes and eigenvectors of different GEDI structural metrics, including the Waveform Structural Complexity Index (WSCI). Relationships between (**b**) WSCI and Foliage Height Diversity (FHD), **c** FHD and canopy height and (**e**) WSCI and canopy height. **d** Distributions of WSCI within canopy height intervals. The two data clusters observed in the PCA biplot correspond to extreme cases of short open forests and tall dense forests. (AGBD = above-ground biomass density, cover = canopy cover fraction, PAI plant area index, RH98 = GEDI canopy height). Dashed lines on (**a**) mark the origin of the PCA space while helping with visual interpretation. The dashed line in (**b**) represents an ordinary least squares (OLS) regression between WSCI and FHD. The dashed lines in (**c**) and (**e**) represent OLS regressions between FHD and log(RH98), and WSCI and log(RH98), respectively.

indirectly related to AGB[28]. High-resolution ALS data are widely available and provide sufficiently detailed measurements to quantify structural complexity accurately and precisely, and therefore represent the best source of training data. Advanced machine learning models can also integrate variables from high-dimensional space to capture patterns beyond traditional ecological metrics like FHD or PAI. Our model-based approach has additionally provided insights into the sensitivity of GEDI waveforms to 3D structural complexity and uncertainty estimates for use in ecological inference, highlighting areas to target for improvement (e.g., tropical savannas).

On the issue of power-law scaling, previous work attributed differences in such scaling across PFTs to crown architecture, vertical layering, and the degree of suppression driven by species competition[31]. Our results expand on these findings globally and suggest consistent evidence of scaling, but one that varies with biome. In particular, the rate of increase of complexity with respect to height was variable by biome, largely driven by differences between PFTs

captured by the different models. However, even within biomes, differences in these scaling patterns at finer scales were observed as the same model can vary its weights (Fig. 2) to predict complexity based on the characteristics of the input canopy profile. Nonetheless, the degree of scaling and its patterns, as given in Fig. 6, are noteworthy.

Our findings distinguished among needleleaf forests, deciduous broadleaf forests, evergreen broadleaf forests, grasslands, shrublands, and woodlands, likely reflecting the different growth strategies adopted by trees in these plant functional types. The broadly different variable importance patterns found between broadleaf and needle leaf forests (Fig. 2) may be attributable to differences in their tree communities and competition for light, affecting the strategies of occupation of the canopy 3D space. For example, needleleaf trees tend to suppress emerging trees in the understory, displaying a more homogeneous canopy structure, dominated by few species able to grow tall enough to occupy the canopy top[32]. Broadleaf forests are generally more diverse in terms of tree species and display a wider array of

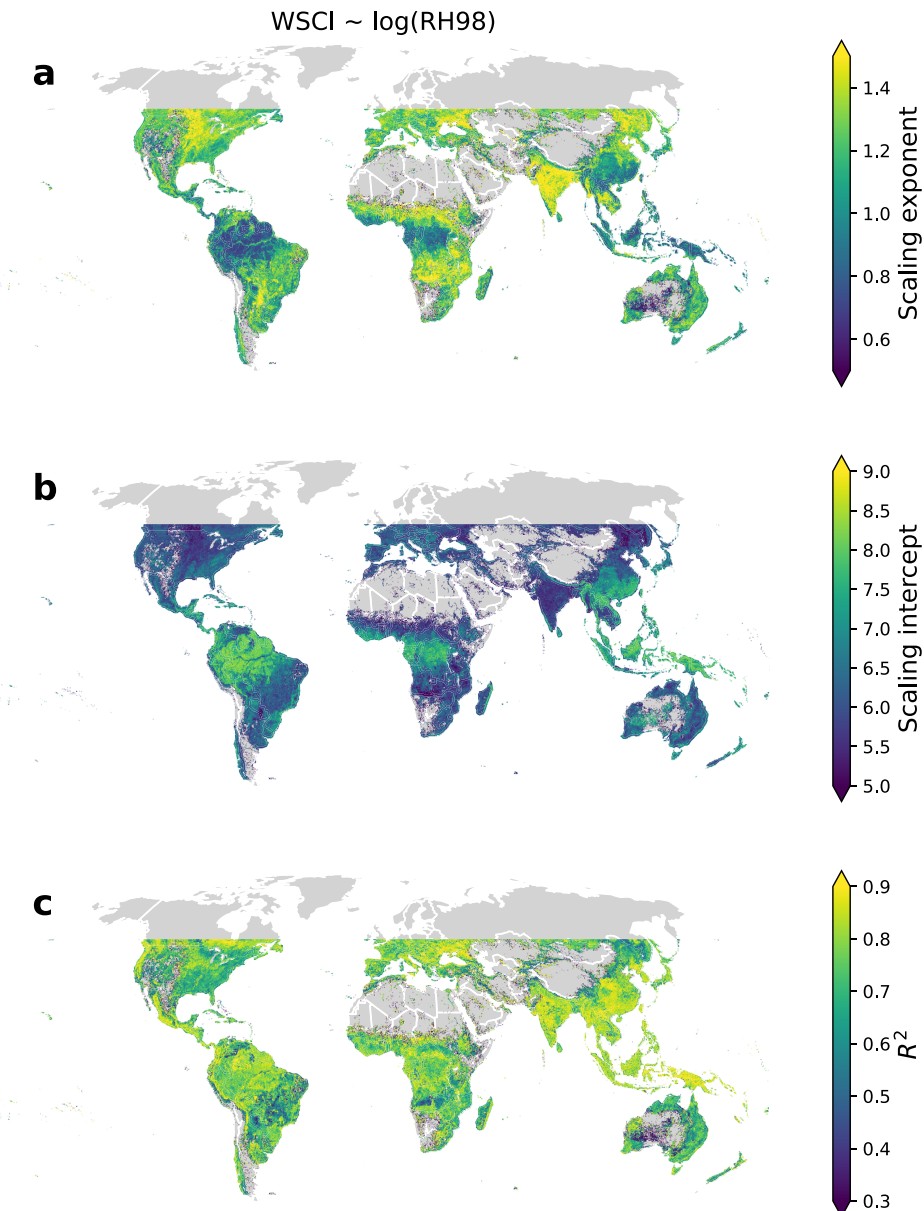

**Fig. 6 | Global relationships between the Waveform Structural Complexity Index (WSCI) and canopy height at 10 km spatial resolution. a** Regression slope representing structural complexity scaling ratio with height, **b** regression intercept, interpreted as the initial state of structural complexity in early seral stages, and (**c**) regression $R^2$, measuring the strength of linear scaling of WSCI with log(RH98). Pixels where a significant statistical relationship was not detected in the ordinary least squares (OLS) linear models ($p > 0.05$ from two-tailed t-tests) were excluded.

growth strategies, enabling more competition to occur among individuals in the understory[31,33]. Forests in GSW have sparse trees mixed with short vegetation of other habits, thus the structural complexity in those areas is partially attributed to low-stature vegetation. However, the GEDI lidar is not optimized to measure the vertical structure of the shortest plants (say less than about 2–3 m) so that GSW model estimates of structural complexity may be dominated by GEDI returns from taller individuals, with low vegetation indistinguishable from the ground. Furthermore, the observed differences in complexity within similar forest types may reflect differences in growth strategies related to variations in species composition and succession dynamics. Tropical forests, generally the most diverse and dynamic ecosystems, benefit from abundant resources supporting tree growth[34,35]. In contrast, other forest biomes with more limited resource availability enable fewer dominant tree species to thrive optimally, which are often under stress induced by competition with individuals following similar

growth strategies seeking the same resources[36]. These properties increase the likelihood of occupying niches in the 3D canopy space in tropical forests, allowing for various resource allocation strategies that collectively boost structural complexity. However, tropical forests across different continents may exhibit distinct mechanisms underlying the development of complexity within the same biome type. For example, African rainforests are known to be less diverse than those in the Amazon[37], particularly in the understory[38]. Conversely, Borneo has a higher proportion of large tree individuals than the Amazon, mainly due to the abundance of species from the Dipterocarpaceae family[39]. Our findings align with these observations, as we found weaker power-law scaling of complexity with height in South/Central American tropical forests compared to African and Southeast Asian tropical forests (Supplementary Fig. 7f). However, although generally positive associations between tree species diversity and structural complexity have been documented[3,5,8,40], the strength of these relationships remains

uncertain. Therefore, the availability of high-resolution WSCI estimates may facilitate further exploration to better understand how biodiversity and structure interact.

The WSCI is derived from structural attributes and is influenced by disturbances underlying GEDI observations. As an effective integrator of structural information (Supplementary Fig. 6), the WSCI may be useful to assess forest degradation and structural integrity (the ecosystem's capacity to maintain its structure, function, and composition relative to its natural range of variation[41]). The $SSCI_{pot}$ product estimates potential structural complexity[16], and its comparison with actual complexity (WSCI) may be informative; for example, large differences between $SSCI_{pot}$ and WSCI where the current complexity of the forest is much less than potential could imply forest degradation and a loss of integrity. However, caution must be exercised when comparing indices that, while designed for the same purpose, may not be equivalent and thus yield unreliable interpretations. In our case here, we found the two indices were only weakly correlated, even when limiting comparisons to intact forests[42] (Supplementary Fig. 9). This lack of agreement is likely indicative of the divergent means by which the indices were derived. The WSCI uses structural attributes directly to model complexity at fine scales (GEDI footprints) spatially across the domain of GEDI observations. In contrast, the $SSCI_{pot}$ relies on climate variables at much coarser scales. Additionally, the $SSCI_{pot}$ models are also influenced by boreal forests, which are not observed by GEDI but are estimated in the $SSCI_{pot}$ map. Given the strong concentration of intact forests in boreal regions it is perhaps not surprising the two indices exhibit a weak correlation (Supplementary Fig. 9). Future research should attempt to derive an equivalent WSCI index from the ICESat-2[43] mission, which has excellent coverage in boreal regions, which if accurate, could provide the means for assessing deviations from potential as given by $SSCI_{pot}$. One limitation of the WSCI is that it is only able to infer horizontal complexity within GEDI footprints through its association with explicitly measured vertical variability within footprints (i.e. the RH metrics). Our results showed that such an association exists, and that horizontal and vertical canopy variabilities are linked. For now, such inferential approaches may be the best that can be hoped for until lidar data with sufficient horizontal resolution and spacing are available over large portions of the Earth. The potential for using high-resolution stereo imagery as a substitute for lidar to resolve canopy features is increasing[44] though its efficacy in dense forests or to measure different vertical portions of the canopy is yet to be determined.

More research is also needed to understand how to use data that are spatially sparse, such as GEDI, to quantify horizontal complexity across spatial scales, linking within and between footprint canopy structure variability. The GEDI data themselves may be used to examine inter-footprint variability in complexity along its sampling transects, as well as focusing on areas of dense coverage that occurred during its mission period where shots were concentrated due to variations in the ISS orbit. Alternatively, structural complexity derived from waveforms, say in 1 km or larger cells, could be examined, but at the cost of losing the ability to examine any fine-scale climatic, edaphic, and disturbance gradients that might exist. Another approach would use multi-sensor fusion to train models using wall-to-wall remote sensing data, say from passive optical or radar data, to infer complexity. For example, Qi et al.[45,46] have shown how GEDI data may be used to train interferometric SAR data from the TanDEM-X satellites to map canopy structure, biomass, and topography at 30 m resolution. Such fusion could be furthered by using extant ALS data sets by way of both calibration and validation. A fusion approach could also help improve the mapping of structural complexity in domains of sparse tree cover where our model estimates over grasslands, shrublands, and woodlands showed weaker performance and higher uncertainty. This result likely reflects the sparse cover in these landscapes and that their structural complexity relies partially on low-stature vegetation not well

observed by GEDI waveforms. Fusion approaches may further enable us to characterize forest structural complexity beyond the GEDI geographical domain[47], e.g. including boreal forests. These could include methods just described as well as potentially leveraging ICESat-2 data.

We have presented the current global pattern of forest structural complexity using a waveform structural complexity index (WSCI) designed to estimate 3D canopy structural complexity from GEDI observations. Acting as an integrator of various structural metrics, the WSCI estimates the contribution of horizontal and vertical canopy elements to predict complexity at any GEDI footprint. Tropical forests consistently exhibited a larger proportion of high structural complexity than other biomes, even for forests of the same height. Our findings further suggest that the canopy relative height metrics that are most important for inferring structural complexity vary across plant functional types and biomes. The creation of a global database of structural complexity from GEDI[48], along with GEDI's explicit measurement of collocated canopy heights enabled us to discover that complexity follows a power law function with respect to height in most forests but with important variations by biome. These high-resolution data were also essential towards confirming that while hotspots of structural complexity are concentrated in tropical forests, some temperate forest sites reached structural complexity levels comparable to tropical forests, and hence actions to additionally preserve these should be accelerated. Such efforts have already begun, notably in the United States, where work is currently underway to identify, monitor, and manage mature and old-growth forests[49]. We anticipate that maps of spatial complexity, such as presented here, will advance these efforts. This latter point emphasizes that much remains to be understood about structural complexity, not only where it occurs, but how it develops, how it may be managed, and what it implies for ecosystem functioning. The widespread availability of WSCI estimates from GEDI is a valuable starting point for developing a quantitative comprehension of these issues.

## Methods

We developed a modeling framework to estimate a 3D structural complexity index calculated from ALS point clouds matched to GEDI footprints and modeled using GEDI RH metrics through XGBoost regression, followed by conformal predictors for estimating model uncertainty. We used SHAP feature explainers[26] to understand the relative contribution of lower, middle, and upper waveform layers for predicting structural complexity in different models and across biomes. We compared worldwide WSCI estimates to other GEDI structural metrics through Principal Components Analysis and subsequent Principal Components Regression to understand how much structural variability captured by GEDI the WSCI was able to uncover. We used linear regression between WSCI and RH98 in 10 km pixels to map geographical patterns of structural complexity scaling with canopy height. Lastly, we compared the global patterns of WSCI with another product estimating potential structural complexity globally to assess whether the two products matched at coarse and fine scales.

### GEDI data

GEDI footprints were filtered to select high-quality, high-sensitivity data (Table 1), guaranteeing that only high-fidelity GEDI footprints were used for model training and validation. Filtering was performed on metrics available in the GEDI L2A version 2 product[19]. Setting a high sensitivity threshold increased the likelihood of keeping only waveforms that reached the ground in dense canopy cover, thus capturing the entire vertical profile of the forest. We further filtered shots based on their Plant Functional Types, keeping only data from PFTs where tree cover is expected based on the MODIS MCD12Q1 V006 product[50]. Our models used relative height (RH) percentiles from the GEDI L2A product as predictors of a 3D structural complexity metric from GEDI intersected Airborne Laser Scanning (ALS) point clouds. Once trained,

**Table 1 | Quality filtering criteria for selecting high fidelity GEDI footprints**

| Filter | Description |
|---|---|
| algorithm_run_flag = 1 | L2B algorithms were applied |
| degrade_flag = 0 | Low degradation of geolocation performance |
| land_cover_data/landsat_water_persistence <10 & land_cover_data/urban_proportion <50 | Non-urban land surface waveforms |
| rx_maxamp > 8 * sd_corrected | Maximum waveform amplitude at least 8x its standard deviation |
| sensitivity > 0.95 (0.98 at the tropics) | High sensitivity shots |
| land_cover_data/pft_class in [1,2,3,4,5,6,11] | Plant Functional Types with tree cover |

**Table 2 | Airborne laser scanning datasets matched to GEDI footprints and used to train the waveform structural complexity index models**

| ALS project | Continental region | Number of intersected GEDI footprints | Reference |
|---|---|---|---|
| CSIR | Africa | 17,284 | Li et al.[76] |
| WWF DRC | Africa | 8713 | Xu et al.[77] |
| JPL Gabon | Africa | 2918 | Fatoyinbo et al.[78] |
| TERN | Australia | 24,185 | Quadros & Keysers[79] |
| G-LiHT Mexico | Central America | 69,880 | Cook et al.[80] |
| PNOA Spain Leon | Europe | 24,020 | Pascual et al.[81] |
| PNOA Spain Extremadura | Europe | 18,245 | Pascual & Guerra-Hernandez[82] |
| JPL Borneo | South East Asia | 4346 | Melendy et al.[83] |
| NERC ARSF Malaysia | South East Asia | 3433 | NERC[84] |
| EBA INPE Brazil | South America | 140,923 | Ometto et al.[85,86] |
| NEON | USA | 502,329 | NEON[87] |

regression models were applied to the current catalog of GEDI footprints (between April 2019 and March 2023) over the Earth's land surface to generate WSCI predictions.

We assessed the relationship between WSCI estimates and other relevant forest structural metrics from GEDI on a global scale, using only shots where RH98 was greater than 5 meters, and where tree cover was expected from the European Space Agency (ESA) WorldCover v200 product at 10 m resolution[51]. The GEDI metrics compared with the WSCI were as follows: canopy cover fraction, canopy height (RH98), plant area index (PAI), foliage height diversity (FHD), and above-ground biomass density (AGBD), extracted from the GEDI L2A[19], L2B[29] and L4A[30] data products.

## Airborne lidar data

We used a comprehensive ALS discrete point cloud database gathered by the GEDI mission team, consisting of datasets shared by research partners and open data initiatives around the world (Table 2). Those point clouds were collected over a multitude of forest sites distributed across five continental regions, covering a broad range of structural (height, cover, PFTs) and environmental (topography, climate) conditions[18].

Those point clouds were matched to GEDI shots collected between 2019 and 2023 and used to correct the systematic geolocation of their intersected GEDI orbit sections (crossovers) using the GEDI simulator framework[52]. This procedure calculated the optimal horizontal and vertical offsets to match an observed GEDI waveform to its ALS-simulated counterpart. Low-quality offsets obtained from less than 10 shots with a Pearson correlation lower than 0.75 were removed from further analyses. Low offset correlations may indicate waveform degradation, not caught by the standard filters, or substantial land cover change in the time period between ALS and GEDI data

acquisitions. Therefore, by filtering out these observations we are guaranteed to keep only consistent ALS/GEDI crossovers over space and time, i.e. registered on the same exact locations with unchanged structural elements over time between ALS and GEDI acquisitions.

## Reference structural complexity index

Several studies have used canopy height variability within lidar point clouds as a structural complexity indicator[21,53,54]. Other forest structural complexity indices are implementations of traditional metrics adapted to lidar data. Foremost among these is Foliage Height Diversity. This metric has long been used in field surveys to summarize the information within forest vertical plant profiles[55] and has a direct lidar counterpart often used as a surrogate for structural complexity[31,56,57]. Other examples of such implementations are fractal box-dimension, the occupation of 3D space independent of scale[22,58]; lacunarity, a measure of structured empty spaces in the vegetation[59,60]; and rugosity and rumple index, proxies for surface roughness[54,61]. More recently, structural complexity indices have been designed specifically for lidar point clouds, addressing complexity as a 3D metric explicitly, and include the Stand Structural Complexity Index (SSCI)[2], the Structural Heterogeneity Index (SHI$_{TLS}$)[23], and the 3D Canopy Entropy Index (CE$_{XYZ}$)[20]. The SSCI combines fractal elements with vertical layering to account for canopy shape, and horizontal and vertical complexity simultaneously, but it was designed for single scan TLS point clouds and lacks information that enables complete 3D coverage since all objects are scanned from a single direction, also lacking penetration of upper canopies due to the lidar instrument's design[14,15,62]. The SHI$_{TLS}$ is calculated by combining explicit tree architectural metrics extracted from Quantitative Structural Models (QSMs), which are digitally reconstructed trees that can only be reliably generated from high resolution and high accuracy multiscan TLS point clouds[15,63]. SHI$_{TLS}$ is a sum of standardized metrics that represent different components of structural variation in the forest canopy, assuming that all metrics have the same weight for explaining the 3D complexity. The CE$_{XYZ}$ is a sensor agnostic index that can be applied to both ALS and TLS point clouds and its conceptualization has an adaptive voxel sampling step to account for varying 3D point densities, which makes it a flexible metric that produces comparable measurements from different lidar instruments and scanning setups. Moreover, the CE$_{XYZ}$ has explicit vertical and horizontal complexity components in its formulation, and similarly to other complexity metrics, it relies on entropy, whose definition from information theory is synergistic with ecological definitions of complexity, as both are closely associated with randomness, heterogeneity, and variability within a system[64]. The SSCI, SHI$_{TLS}$, and CE$_{XYZ}$ have a strong theoretical basis on forest ecology and forest management principles, and all have been empirically validated across forest stands from different ecoregions and under different management regimes[2,20,23].

Our models targeted the 3D Canopy Entropy Index (CE$_{XYZ}$)[20] to take advantage of the large training database of paired ALS/GEDI crossovers. Although indices designed for TLS data are desirable as a modeling basis due to their higher level of structural detail, the substantially lower coverage and availability of TLS data makes it harder to have a sufficiently large database of collocated TLS/GEDI crossovers

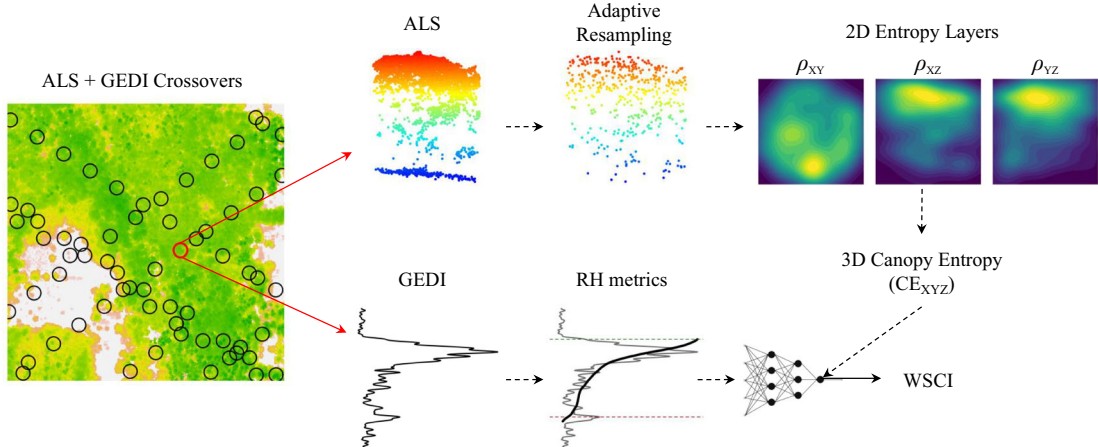

**Fig. 7 | GEDI Waveform Structural Complexity Index (WSCI) modeling framework.** Data processing pipeline to model 3D Canopy Entropy ($CE_{XYZ}$) as a function of GEDI relative height percentiles (RH metrics) from Airborne Laser Scanning (ALS) point clouds and GEDI footprints crossovers. ($\rho_{ij}$ = probability density planes).

for training models at a global scale that captures sufficient variability in most ecoregions. Moreover, there are currently no GEDI waveform simulation frameworks validated for TLS point clouds, and therefore no reliable way of performing geolocation matching and upsampling using simulated waveforms from TLS data.

The $CE_{XYZ}$ combines entropy measures from 2D probability density planes, estimated by a kernel density function, measuring the amount of information contained in a 3D point cloud scene (Eq. 1). Liu et al.[20] demonstrated that $CE_{XYZ}$ translates well to forest structural complexity by testing it under multiple forest conditions, showing a monotonic growth of the index with increasing vertical layering and tree density. Furthermore, the $CE_{XYZ}$ is defined as a function of explicit components of horizontal and vertical canopy entropy (i.e. complexity), enabling us to measure the relative contribution of these components in the WSCI modeling framework.

$$CE_{XYZ} = \sqrt{CE_{XY}^2 + CE_{XZ}^2 + CE_{YZ}^2} \qquad (1)$$

$$CE_{2d} = -\sum_{i=1}^{n} \rho_i \cdot \ln(\rho_i) \cdot s^2 \qquad (2)$$

Where $CE_{XYZ}$ = 3D Canopy Entropy, $CE_{2d}$ = Canopy Entropy of a 2D plane (XY, XZ or YZ), $\rho$ = estimated probability density kernel of a point $i$ on the 2D plane, $s$ = grid size (regular spacing to measure kernel density, in point cloud units).

We measured $CE_{XYZ}$ from the point clouds in the ALS/GEDI crossovers, which corresponded to the total 3D entropy of 25 m diameter circular plots, with kernel density calculated in systematic grids of 10 cm spacing. We then modeled $CE_{XYZ}$ as a function of the RH metrics from the GEDI footprints (Fig. 7). Point cloud processing and $CE_{XYZ}$ calculations were done in the R programing language using the packages lidR[65] version 4.1.0, TreeLS[66] version 2.0.5, ks[67] version 1.13.5, sf[68] version 1.0.12, nabor[69] version 0.5 and trend[70] version 1.1.4.

**Modeling framework, feature importance, and uncertainty estimation**

The high-quality crossovers were split into training and test data sets on a geographical basis, in which crossovers from 80% of the sites (training set) were used for training regression models and the remaining 20% (test set) were used for estimating model prediction intervals. The WSCI models were fitted using the Extreme Gradient Boosting Trees (XGBoost[25] Python package version 1.7.4) regression algorithm, as it offers a good balance of computational efficiency, precision, accuracy, and robustness against overfitting[25]. These

XGBoost properties granted us fast processing time with reliable outputs, which enabled us to carry out extensive hyperparameter tuning for better model optimization and apply strict quality control on model selection. We used GEDI RH metrics as predictors in the XGBoost regression models, as the vector of RH metrics for a single GEDI footprint has a fixed length of 101 (0 to 100%) and retrieves information from the full forest vertical profile. Therefore, we expected little information loss relative to training models using the raw GEDI full waveforms, with the added benefit of significantly faster computations on both model fitting and prediction. Global models were fitted for each forest PFT domain, based on GEDI footprint intersections with pixels from the MODIS MCD12Q1 V006 product[50].

Hyperparameter tuning was performed through 5-fold grid search with spatial cross-validation (GridSearchCV) using the scikit-learn[71] Python package version 1.3.2, minimizing the average root mean squared error (RMSE) cost function over the training set split into folds containing samples from different locations. The hyperparameters optimized by the GridSearchCV were: (1) number of estimators (regression trees), (2) sub-sample size (observations), (3) feature space sample size (variables), (4) maximum regression tree depth and (5) learning rate. Spatial cross-validation folds were split on a geographical basis, thus the data in the validation fold belonged to geographical locations unseen in the folds used for fitting the model on each iteration of the GridSearchCV. To enforce model and geographical generalization, we picked the optimal hyperparameters from models that minimized RMSE on validation folds while maintaining a difference of less than 5% in RMSE and $R^2$ between training and validation folds.

We used SHAP explainers[26] to extract feature importance from model predictions. SHAP explainers represent the average marginal contribution of a feature to all possible subsets of features using the shap[26] Python package version 0.43. It is an approach derived from game theory that is generalizable to any type of machine learning model, providing a comprehensive and consistent way to understand the impact of individual features on model predictions. We used SHAP explainers on the predictions from all the training GEDI samples to understand the patterns of feature importance across different PFT models. We also extracted feature importance from all GEDI high-quality observations to map the geographical patterns of relative contributions of canopy vertical strata to model predictions worldwide. We defined three waveform layers: lower, as the accumulated feature <= RH33; middle, as the accumulated feature importance > RH33 and <= RH66; and upper, as the accumulated feature importance > RH66. We averaged the canopy strata contributions in 10 km pixels

and generated an RGB false color composite image representing the feature importance from these three canopy strata in different bands.

We used conformal predictors to calculate prediction intervals in the WSCI models through the crepes[72] Python package version 0.6.1. This technique allows the estimation of prediction intervals around individual predictions estimated by a known model[73,74], being able to build uncertainty estimators using any kind of pre-trained machine learning model as input, accompanied by a set of observations unseen during model training (our test set), used to calibrate the error model. We trained conformal predictors in Mondrian intervals[74] to account for heteroskedasticity in model residuals at different intervals of the modeled variable. WSCI predictions were calculated for the entire GEDI footprint catalog acquired between April 2019 and March 2023, providing complexity estimates accompanied by model prediction intervals for every GEDI footprint over the Earth's land surface.

## Horizontal and vertical complexity model contributions

The WSCI models were fitted to predict $CE_{XYZ}$ directly from the GEDI RH metrics, but to assess the amount of structural complexity information explained across in the vertical and horizontal directions within GEDI footprints we also trained XGBoost models to predict the different terms defined from the $CE_{XYZ}$ formula (Eq. 1). We used the same modeling framework described above to estimate horizontal ($CE_{XY}$) and vertical ($CE_{Z}$) complexity from GEDI RH metrics on a PFT basis. Since GEDI only measures vertical complexity in a single direction, we defined $CE_{Z}$ as the average between the two vertical planes entropy in Eq. (1): $CE_z = \frac{CE_{XZ} + CE_{YZ}}{2}$. We assessed model performance through the RMSE and $R^2$ statistics to determine the efficacy of GEDI for explaining the variability in structural complexity in both directions. We also assessed the feature importance of each model through SHAP explainers to determine which RH metrics are better predictors of horizontal and vertical complexity.

## Mapping global patterns and relationships with other GEDI metrics

We generated a global map of the mean WSCI and mean model prediction interval at 1 km spatial resolution. To remove non-forest observations, we aggregated only observations where RH98 was greater than 5 meters, and where tree cover was expected from the ESA WorldCover v200 product at 10 m resolution[51].

We compared WSCI estimates with canopy cover fraction, RH98, PAI, FHD, and AGBD, extracted from the GEDI L2A[19], L2B[29], and L4A[30] data products. A PCA was performed on a random sample of 1% of the forest footprints used for mapping. All metrics were standardized to the [0,1] range before performing the PCA to avoid unequal weighting of the PCA eigenvectors caused by different scales of measurement units from the different variables. We also performed a PCA excluding WSCI as input and then performed a PCR to predict WSCI using the first three PCA components to quantify the structural variability uncovered by the WSCI relative to the combination of other GEDI high-level metrics. Ordinary least squares (OLS) regressions were subsequently carried out to investigate relationships between WSCI and log(RH98), used as a proxy of canopy height, in 10 km pixels for the entire GEDI domain. Both PCA and OLS analyses were performed in the Python programing language using packages scikit-learn[71] version 1.3.2 and statsmodels[75] version 0.14. To capture biome-wide trends and reduce outlier effects in the relationship between WSCI and canopy height we used robust linear regression, fitting an iteratively reweighted least squares regression algorithm weighted using Tukey's biweight function in the statsmodels[75] Python package version 0.14. We applied robust regression between WSCI and log of RH98 to the 1% GEDI footprints sampled for the PCA, divided into 3 forest biome categories: tropical, temperate or other forest domains, using the WWF terrestrial ecoregions of the world as basis[24].

## Reporting summary

Further information on research design is available in the Nature Portfolio Reporting Summary linked to this article.

## Data availability

GEDI data are openly available and archived on NASA Distributed Active Archive Centers (DAACs). The Waveform Structural Complexity Index (WSCI) data product is openly available in the Oak Ridge National Laboratory (ORNL) DAAC as GEDI04_C Waveform Structural Complexity Index Product under accession code https://doi.org/10.3334/ORNLDAAC/2338. All Airborne Laser Scanning (ALS) datasets used in this study are available under open access and can be obtained directly from the sources or by contacting the principal investigators listed in Table 2 of this study. The GEDI footprint-level RH metrics used to train the WSCI models were taken from the GEDI02_A height and elevation product, available at the Land Processes (LP) DAAC under accession code https://doi.org/10.5067/GEDI/GEDI02_A.002. GEDI's FHD, PAI and cover metrics were taken from the GEDI02_B canopy cover and vertical profile metrics product also available at the LP DAAC under accession code https://doi.org/10.5067/GEDI/GEDI02_B.002. GEDI's footprint-level biomass data were taken from the GEDI04_A aboveground biomass density (AGBD) product, available at the ORNL DAAC under accession code https://doi.org/10.3334/ORNLDAAC/2056. The ESA worldcover v200 data product is available at https://worldcover2021.esa.int. The WWF Terrestrial Ecoregions of the World can be obtained at https://www.worldwildlife.org/publications/terrestrial-ecoregions-of-the-world.

## Code availability

The code used to generate training data from ALS and apply the WSCI models to GEDI data is available on GitHub at https://github.com/tiagodc/GEDI-WSCI. A permanent reference to the version of the code used in this study is available at https://doi.org/10.5281/zenodo.13351657.

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

## Acknowledgements

The authors gratefully acknowledge the large number of contributors of airborne lidar data that enabled the creation of the empirical WSCI models described in this study. Access to the CSIR Africa ALS data was under a data use agreement between the GEDI Mission Science Team data and we gratefully acknowledge CSIR and the University of Witwatersrand for funding the collection of these data. We thank Adrian Pascual and Matheus Nunes for their support on early quality assessments of the WSCI product and manuscript. We are indebted to Bryan Blair and Michelle Hofton for their invaluable contributions to waveform processing and entropy exploration, which laid the groundwork for this research. We gratefully acknowledge the funding from National Aeronautics and Space Administration (NASA) contract NNL 15AA03C for the development and execution of the GEDI mission, including funding to R.D. and J.A., and NASA FINNEST grant 80NSSC22K1543 to T.C.

## Author contributions

Conceptualization: T.C., R.D. Methodology: T.C., J.A., R.D. Investigation: T.C., J.A., R.D. Visualization: T.C. Funding acquisition: R.D. Writing - original draft: T.C., R.D. Writing - review & editing: T.C., J.A., R.D.

## Competing interests

The authors declare no competing interests.
