## [Peer Review File · Nature Communications]

Characterizing the structural complexity of the Earth's Forests with spaceborne lidarREVIEWER COMMENTS

Reviewer #1 (Remarks to the Author):

This study applies an index of structural complexity derived for ALS to incorporate a modeled horizontal component of structural complexity to GEDI vertical waveforms (RHs). The resulting index is then applied globally (though this isn't a map) to assess which vertical elements of the GEDI waveform are driving structural complexity across biomes. The authors assess power law relationships between this complexity index and other prominent indicators of canopy structure, before comparing it to a coarse global model that ostensibly models potential complexity based on a few highly clustered TLS samples. The WSCI is novel, processing and analytical steps are mostly robust, the manuscript is well written, and the paper has the potential to be highly valuable to the community if current weaknesses (in order of major, minor, and row-by-row critiques) can be adequately addressed.

My major issues with the manuscript are as follows:

1. The comparison with SSCIpot is distracting and ultimately diminishes the quality of the paper. I would strongly suggest dropping this component of the paper. SSCIpot is derived from a small number of fine-scale plots (TLS), greatly clustered in 20 primary forest sites, yet is ambitiously applied to the globe based on their relation to environmental layers. That study found "global variation of forest structural complexity [to be] largely explained by annual precipitation and precipitation seasonality ($R^2 = 0.89$)". Allow me to suspend my disbelief (in light of years of experience doing similar work) that coarse precipitation layers so accurately describe what is an incredibly complex process/outcome of community assembly (resource acquisition, stress, competition, evolutionary history, disturbance, etc.) and assume the relationship somehow holds globally. SSCIpot is not a map, and is derived by modeling the few hundred TLS scans to coarse environmental layers to produce a global SSCIpot, that is not formally validated, and ultimately speculative. To employ the old adage that applies to many of our best attempts to describe nature through data and models: "garbage in garbage out". But if there were some insight gained from comparing WSCI with an unvalidated and unclear global map it'd likely be worth it, no? Well, the authors find similar patterns, albeit with some fine scale deviations. The way I read this, is that there's not much to see here. So what was gained? Little to nothing in my view. What was lost? In my opinion, a lot. The digression distracts the reader from the main points of the analysis while providing little in the way of insight, and furthermore diminishes readers' confidence in the results if the chief comparison is a speculative, unvalidated product based on coarse environmental layers. Perhaps if SSCIpot was GEDI derived, I could buy its relevance to the current analysis (at least it'd have a large sample size and an even sample distribution). Alas, no again. Instead, what appears to drive SSCIpot is in fact its underlying environmental layers. If so, why not just do that explicitly? And using better environmental data while you're at it (I'm not suggesting you do that in this manuscript, btw). I think this manuscript is robust and stands on its own without it, and importantly, is diminished by including it.

2. While the researchers note that GEDI waveforms do not provide information on horizontal

structure, I find it a little disappointing that WSCI is not GEDI based but highly derived via secondary models with ALS-based horizontal complexity. This is likely unavoidable and I still commend the effort, but I think it should be made more explicit that WSCI is not a GEDI index per se. Ultimately, it's a convoluted way to mine data from the same waveforms other researchers are assessing. The authors could better address what WSCI actually means in the context of GEDI if it needs to be modeled from ALS and it's implications for deriving global GEDI indices (Disc).

Minor points:

1. The researchers employ GEDI RHs as the fundamental predictor. While RH has greater fidelity to the original waveform, it suffers from light extinction / signal saturation. Perhaps I missed this, but were these factors accounted for? Were RHs otherwise normalized or transformed? How were negatives dealt with?
2. Bullseye co-location still has associated sigma. How was this accounted for when co-locating GEDI footprints with ALS?
3. At several points (eg 126-129; 263; 204-211), the researchers make mention of variable importance via SHAP values, but did not provide these results. Instead, I'm squinting at tiny distributions on the figure margins (eg Fig. 1). It would be nice to see an analytical paper trail of sorts. This could be a good Supp table/figure and would support findings on how low-mid-upper canopy layers contribute to different complexity profiles in tropical versus temperate forests.
4. I'm not sure how much the "lost" explained variance in the PCA models is due to WSCI (See my comments corresponding to row 247). I see alternative explanations for the same phenomenon, and think there are other ways, less susceptible to error propagation, to construct a null model to test that hypothesis.
5. The authors repeatedly refer to this as fine resolution. Well, maybe GEDI footprints count (though certainly not in comparison to ALS or TLS), but the subsequent analyses are clearly not. Nor are they even landscape scale. There are clear strengths to this analysis, but fine resolution is not one of them. Please try to clarify, and in the process, I would suggest addressing an obvious strength of the analysis: consistency over massive extents, two things TLS and ALS decidedly lack.

Line-by-line comments:

25 - 3D "forest" or "canopy" complexity

26 - I read the results to say not that tropical forests are the "most" complex (which I interpret as having the greatest levels of complexity as measured by WSCI) but instead were, on net, "more" complex. This is a small but important point. While the median tropical forest is more complex than the median temperate one, temperate forests are still capable of the highest levels of complexity on par with those found in the tropics. For example, how do those Bornean Diptocarps compare with

Redwoods? I imagine they are commensurate, and as WSCI and FHD are highly dependent on canopy height, I wouldn't be surprised if Redwoods had the world's highest levels of complexity ("most complex"), as they rank in biomass.

31 - I wouldn't call these spatial distributions "nuanced". First, I am not clear what a nuanced distribution is. Do you mean they feature finer resolution, greater precision, or more fine-grained patterns in spatial turnover? Elucidating on any of these characteristics didn't seem to me to be a goal of this paper. Instead, I think the "nuance" (not my word) comes from GEDI characterizations of vertical canopy profiles, which in this case likewise has a modeled horizontal component.

42 - Neither of those studies dealt with ecosystem function explicitly. Perhaps "biodiversity" is the word. Further, among the two, I would suggest substituting the EA paper for one recently published by the same lead in Environmental Research Ecology, which unlike those two, is an explicit test of GEDI for characterizing patterns in biodiversity (also sees rows 263 and 370).

63 - Also, TLS signal saturation for larger/taller canopies

74-78 - See major point #1. Here and below.

88 - "empirical"

104 - Earlier you established the state of the field from TLS and ALS studies, so I wonder if "high-resolution" is the best way to distinguish your advancements in processing GEDI waveforms.

105 - Did this study assess how "high-resolution estimates" of complexity "result from gradients in environmental drivers, seral stage and disturbance conditions." If not, I would either couch this sentence in terms of potential implications of the results and/or cite another study that does touch on these topics.

124 - Linking to Supp. 1; surprising to see bias of 0.00 (versus -0.00, lol). In fact eyeballing the charts seems to definitely show an, acceptable IMO, bias. Can you confirm that these bias estimates are correct?

126-129 - Where are these results? Fig 1 insets are too small to assess these relationships (for a reader). This could perhaps be a good Supp figure.

115 - While you go into more detail in the Methods, I think a passing mention to the fact that GEDI shots are co-located with ALS would make your process more clear to the reader at this point.

147 - Why 6.3? Is this an artifact of the CExyz equations? I think this text, and any subsequent explanations, would be better located in the body text, not the caption.

153 - Linking to Supp. 2; why is the Amazon Basin a continuous polygon and the other locations, more or less, discrete points?

180 - High quality (eg filtered) shots, no? Also, small typo: "shots and averaged"

204-211 - Here again, it would be nice to see an analytical paper trail of some sort (see minor points #3). Otherwise, I'm squinting at tiny distributions on the figure margins.

225 - "canopy structure"

229 - How were these samples? Random? Stratified?

232 - Be explicit with PCA1 interpretation like you are with PCA2. Perhaps start off by saying more explicitly that "density" is driving PCA1 (if that's your interpretation).

236 - And/or height? FHD is correlated with height, to a large degree because it is highly determined by it. Half of Shannon's H is "S", which in this case is the total number of height bins, from which relative proportions are summed.

244 - Not a question, so no "?" needed.

247 - I'm not sure how much of this "lost" explained variance is due to WSCI, but rather the imperfect nature of linear models to capture variance in the data. Put another way, while PCA1-3 explains 99.5% (seems high, but okay I won't quibble) how much would it explain if you inverted it (PCA still retaining the WSCI information) to predict WSCI? I imagine somewhere in between 85-99.5% and likely towards the former. In other words, could you come to the same conclusion with WSCI retained.

250 - What assumptions are you referring to? If light extinction / signal saturation is one, it could be a good assumption to retain.

254-260 - Capitalize acronyms

290 - Extra comma

297-308 - I would suggest dropping this section. Its underlying assumptions are not robust (comparing your gridded maps to an unvalidated highly speculative and likely problematic secondary map based on limited ground samples). Further, the results aren't exactly compelling: similar patterns, with some fine scale deviations. I do not finish this section feeling like I learned something about the world like I did in other sections.

306 - Extra comma

307 - Missing an "as" before "well as"

308 - But GEDI is not swath mapping and has huge gaps in the tropics, so is this really fine-grained

and local? No. But there are other advantages to GEDI ...

326 - How much of this effect is driven by biome versus forest type (eg conifer and broadleaf)?

329 - "6,"

335 - You hit resource abundance pretty hard, but might mention two other important axes hypothesized to drive community dynamics a la Grime: stress, and most importantly, competition.

345 - Does it though? There seem to be many studies that almost universally find consistent positive relationships. Also, there's an extra "." here.

348-352 - Yes! Is this actually just a comparison to coarse environmental layers? And if so, why not just do that explicitly, and using better environmental data while you're at it! (I'm not suggesting you do that in this manuscript, btw)

363 - Very glad you note this explicitly here. And, IMHO, it could be a more prominent point in earlier section (even abstract) without distracting from the strength of your results.

376 - Sparse tree cover, yes. But also sparse GEDI shot distributions.

460 - "sensor agnostic index" is important enough of a point that you may consider mentioning it in the body text.

470 - Redundant from paragraph before

503 - Were RHs normalized or transformed? Was anything done to address negative values?

556- PCR? Did you mean "PCA"?

Reviewer #2 (Remarks to the Author):

The authors present a newly developed forest structural complexity index (WSCI) based on GEDI spaceborne LiDAR data. This index is conceptually based on a recently published structural complexity index derived from airborne laser scanning data (CExyz by Liu et al. 2022) and applied to (nearly) globally available GEDI data to produce a global map of forest structural complexity. This dataset is then used to (1) compare WSCI with other GEDI derived structural metrics, (2) test scaling relationships between canopy height and structural complexity, (3) to compare the near-global forest structural complexity map with a previously published map by Ehbrecht et al. 2021, and (4) to compare the structural complexity of forests of different plant functional types (PFT) and how these are driven by structural differences in their under-, mid-, and overstory.

The manuscript is generally very well-written and addresses a timely and relevant topic. My main

concerns are related to the methodological approach, some conclusions drawn from the results and whether the newly developed index WSCI provides additional information on top of what other GEDI derived structural indices provide already. Furthermore, I think that the manuscript requires more clarity regarding the terms used and how the authors define structural complexity. I will try to outline my concerns in more detail in the following and hope to provide some ideas on how the manuscript could be improved.

- The overall motivation and rationale of the study is clear. However, the introduction lacks a clear definition of what the authors are actually aiming to measure with WSCI. In line 356-357, the authors state that “there is no established definition of structural complexity so that the concept of a “true” complexity is fraught”. While I generally agree that a commonly accepted definition of structural complexity is lacking, there have been some attempts to define structural complexity that I think should be acknowledged here. In their review on relationships between plant diversity and structural complexity, Coverdale & Davies (2023, <https://doi.org/10.1111/1365-2745.14068>) discuss current definitions of structural complexity (see chapter 2 of their paper). The reviewed definitions by Ishii et al. (2004), Ehbrecht et al. (2021), Gough et al. (2019) and Atkins et al. (2018) all tend to point into similar directions, i.e. defining structural complexity as variability/heterogeneity of biomass/canopy element distribution in 3D space. Moreover, Ehbrecht et al. (2021) provided a theoretical framework on the drivers of structural complexity (Fig. 1) and LaRue et al. (2023) presented a theoretical framework for the ecological role of structural complexity. I think the introduction would strongly benefit from taking these definitions and state of research in this field into account.

- The study’s main objective is the development of a footprint-scale Waveform Structural Complexity Index. What I think is lacking here are research questions or hypotheses. To improve the manuscript, I’d suggest to formulate research questions or hypotheses that are then being addressed in the results and discussion section.

- I do not fully understand how horizontal complexity can be derived from GEDI footprint data, when GEDI does not measure horizontal variability on footprint level, as the authors mention in line 120-121? Furthermore, I appreciate if the authors could present in more detail the difference between horizontal and vertical complexity and how both contribute to overall 3D complexity? I think it becomes surely clearer when looking at Liu et al.’s paper where the CExyz index is being introduced. However, to make the presented study more intuitively understandable, it should be clarified here without referring to Liu et al.’s paper.

- The accumulated feature importance was divided into under- (\leq RH40), mid- ($>$ RH40 and \leq RH90), and overstory ($>$ RH90) layers. The selection of the respective RH values as breakpoints for the different layers strongly affects the results. What was the basis for the selection? Was the classification based on an approach presented in another study? Other studies, e.g. Willim et al. 2020 (<https://doi.org/10.3390/rs12121907>) used other classifications (e.g. RH33, RH66 and $>$ RH66). Using different breakpoints for classification would likely yield different results and thus probably result in drawing different conclusions.

-The authors show that WSCI correlates well with other structural metrics, especially RH98 and and Foliage Height Diversity, and particularly ask the question whether WSCI “captures any variation in complexity that is not already captured with existing standard GEDI metrics”. If 3 principal components from the PCA explain 85% of the variance in WSCI, can we confidently conclude that WSCI “captures variation not included in these other metrics”? My question is, would we come up with different conclusions about the global patterns of structural complexity if we simply used FHD? Would WSCI allow us to identify differences between forest types that we wouldn't see by using FHD?

-A few thoughts about index construction (figure 5a): PC1 (with Cover and PAI) quantifies something like a density component, while PC2 (with WSCI, FHD and RH98) quantifies something like a vertical structure component. Other indices, like e.g. SSCI, are also composed of a kind of density component (perimeter-area-ratio of polygons) and vertical structure (ENL, which is basically the same as FHD, but with a different scaling ($\exp(\text{FHD})$). See also line 38-39. Wouldn't WSCI rather provide added information and be more complementary to other existing indices (i.e. more different from Rh98/FHD), if it was based on a combination of Cover/PAI and FHD/RH98. In the PCA, it would then probably load on the axis between Cover/PAI and FHD/RH98. I am particularly wondering if the authors have tried or experimented with different approaches for index construction. If so, it would be great if these could also be presented in the SI.

-Comparing the actual structural complexity as measured by WSCI to the potential structural complexity measured by globally modelled SSCI data is very interesting. Despite a low R^2 , I find it quite surprising that WSCI and SSCIpot show a significant correlation, given the fact that (1) the two indices quantify structural complexity from completely different perspectives based on completely different approaches, and (2) because SSCIpot quantifies the level of structural complexity that is possible in the absence of disturbances. Against the background that the largest part of the world's forests show signs of anthropogenic disturbance, wouldn't it make sense to (additionally) focus on a comparison of WSCI and SSCIpot for old-growth or primary forests? Generally, I am concerned about drawing conclusions from a global comparison -as the authors themselves also point out in line 354- given the fact that both indices are very different by design. The comparison could be more straightforward, if WSCI had a design similar to SSCI (see also my earlier comment on index construction). Thus, additionally focusing a comparison on primary forests could potentially eliminate a few confounding factors. When discussing the results, I also suggest to put more emphasize on the potential of comparing actual and potential complexity to quantify forest structural integrity or degradation, and outline which further work is needed to facilitate proper comparisons.

I enjoyed reading and reviewing the manuscript. I hope my comments may help to improve the paper.

Kind regards
Martin Ehbrecht

Reviewer #3 (Remarks to the Author):

Dear Authors,

I really enjoyed reading your manuscript and think you conducted a state-of-the-art study to analyze the structural complexity of forests globally. I think it is a valuable demonstration on how to turn technical data into useful environmental information and their interpretation.

Here are some (minor) thoughts and suggestions about the manuscript that I think are worth including:

1. Readers that are unfamiliar with the GEDI system might wonder, why all the maps show no data around the poles and why you did not include boreal forests in your analysis. I suggest adding the technical information about GEDI data availability in the introduction and briefly discuss the implications of excluding boreal forests in your research.
2. Usually Light detection and ranging is abbreviated as LiDAR, I personally don't mind the all lower case lidar but some readers might.
3. From a modelling point of view, one thing you should add and discuss in the manuscript is the impact of the distribution of the ALS training data. Why is your model performance not as good in regions outside the tropics (as seen in Figure 3b and Supp. Figure 2)? Why is it especially low in the western USA - despite the fact that this is a region the model should be quite good because there are ALS data available? And what implications does this error distribution have for regions with no ALS data available (e.g. Europe or Central Asia)? In other words, is your model trustful in these areas? I think for a better understanding and writing of this point, it could be worth it to move Supp. Figure 2 into the main manuscript or show a small ALS distribution map in Figure 3.
4. Two random spelling errors I encountered during reading: l88 - empirically -> empirical; l116 - database -> data

Best regards
Marvin Ludwig

REVIEWER COMMENTS

Reviewer #1 (Remarks to the Author):

This study applies an index of structural complexity derived for ALS to incorporate a modeled horizontal component of structural complexity to GEDI vertical waveforms (RHs). The resulting index is then applied globally (though this isn't a map) to assess which vertical elements of the GEDI waveform are driving structural complexity across biomes. The authors assess power law relationships between this complexity index and other prominent indicators of canopy structure, before comparing it to a coarse global model that ostensibly models potential complexity based on a few highly clustered TLS samples. The WSCI is novel, processing and analytical steps are mostly robust, the manuscript is well written, and the paper has the potential to be highly valuable to the community if current weaknesses (in order of major, minor, and row-by-row critiques) can be adequately addressed.

We thank the reviewer for his careful review of our manuscript and his appreciation of our efforts. We have tried address all of the concerns as listed below.

My major issues with the manuscript are as follows:

1. The comparison with SSCIpot is distracting and ultimately diminishes the quality of the paper. I would strongly suggest dropping this component of the paper. SSCIpot is derived from a small number of fine-scale plots (TLS), greatly clustered in 20 primary forest sites, yet is ambitiously applied to the globe based on their relation to environmental layers. That study found "global variation of forest structural complexity [to be] largely explained by annual precipitation and precipitation seasonality ($R^2 = 0.89$)". Allow me to suspend my disbelief (in light of years of experience doing similar work) that coarse precipitation layers so accurately describe what is an incredibly complex process/outcome of community assembly (resource acquisition, stress, competition, evolutionary history, disturbance, etc.) and assume the relationship somehow holds globally. SSCIpot is not a map, and is derived by modeling the few hundred TLS scans to coarse environmental layers to produce a global SSCIpot, that is not formally validated, and ultimately speculative. To employ the old adage that applies to many of our best attempts to describe nature through data and models: "garbage in garbage out". But if there were some insight gained from comparing WSCI with an unvalidated and unclear global map it'd likely be worth it, no? Well, the authors find similar patterns, albeit with some fine scale deviations. The way I read this, is that there's not much to see here. So what was gained? Little to nothing in my view. What was lost? In my opinion, a lot. The digression distracts the reader from the main points of the analysis while providing little in the way of insight, and furthermore diminishes readers' confidence in the results if the chief comparison is a speculative, unvalidated product based on coarse environmental layers. Perhaps if SSCIpot was GEDI derived, I could buy its relevance to the current analysis (at least it'd have a large sample size and an even sample distribution). Alas, no again. Instead, what appears to drive SSCIpot is in fact its underlying environmental layers. If so, why not just do that explicitly? And using better environmental data while you're at it (I'm not suggesting you do that in this manuscript, btw). I think this manuscript is robust and stands on its own without it, and importantly, is diminished by including it.

We modified our results to address these concerns. We agree that the results from a comparison between WSCI and SSCIpot were a digression from the main results of the paper, thus we dropped this section from the results. However, the SSCIpot work was an early attempt to map structural complexity globally and also published in *Nature Communications*, therefore its relationship to our work should be acknowledged. We now elaborate on SSCIpot limitations in the Introduction (line 93) and provide more constrained analysis in supplementary information, following Reviewer's 2 suggestions, by comparing

only intact forests for each product, and discuss issues of comparability between WSCI and SSCIPot in the discussion (line 433).

2. While the researchers note that GEDI waveforms do not provide information on horizontal structure, I find it a little disappointing that WSCI is not GEDI based but highly derived via secondary models with ALS-based horizontal complexity. This is likely unavoidable and I still commend the effort, but I think it should be made more explicit that WSCI is not a GEDI index per se. Ultimately, it's a convoluted way to mine data from the same waveforms other researchers are assessing. The authors could better address what WSCI actually means in the context of GEDI if it needs to be modeled from ALS and its implications for deriving global GEDI indices (Disc).

We apologize for any confusion on the derivation of WSCI in the first draft. We have used clearer language when describing WSCI as a modeled product in the introduction:

- Line 118: This index is established by modeling the empirical relationships between GEDI RH metrics and an existing metric of 3D structural complexity built upon explicit measures of horizontal and vertical complexity (CE_{xyz})²⁰ derived from ALS point clouds.

We also discussed more in depth the rationale behind our modeling approach and its benefits:

- Line 393: Structural complexity is inherently linked to 3D space^{1,2,4}, hence our choice of a metric that separates complexity into vertical and horizontal components²⁰. Metrics such as FHD simply ignore the horizontal component. GEDI waveforms only provide direct measurement of the vertical component, leaving the horizontal component to be inferred indirectly through our model-based approach. This approach is analogous to the modeling of above ground biomass (AGB), which relies on in situ plot measurements to train prediction models based on measurements (e.g. height) that are indirectly related to AGB³¹. High-resolution ALS data are widely available and provide sufficiently detailed measurements to quantify structural complexity accurately and precisely, and therefore represent the best source of training data. Advanced machine learning models can also integrate variables from high-dimensional space to capture patterns beyond traditional ecological metrics like FHD or PAI. Our model-based approach has additionally provided insights into the sensitivity of GEDI waveforms to 3D structural complexity and uncertainty estimates for use in ecological inference, highlighting areas to target for improvement (e.g., tropical savannas).

To expand on the above, ALS, along with TLS is essentially considered ground truth for structure. If we want to get at some understanding on the role of horizontal complexity within canopies, it is, as the reviewer states, largely unavoidable that we must model that component using ALS. We consider this an opportunity, and not a limitation, given that the GEDI mission has developed what is perhaps the most comprehensive validation data base of field and coincident ALS measurements yet compiled. The use of models also has the benefit of helping understand what portions of the waveform are important for predicting complexity and how these vary by PFT and region.

It is further helpful to consider the future direction of ecosystem characterization which will certainly be dominated by lidar. One example is the recent GEO-TREES initiative (<https://geo-trees.org/>). GEO-TREES is an expansive, international effort to provide a high-quality set of reference ground plots, along with detailed canopy characterization using ALS and TLS, mainly for improving biomass estimates from space missions, but also to understand the multifaceted aspects of structure, function and composition. Our approach provides a direct means of comparing structural complexity derived using ALS/TLS over plots from such initiatives with those derived from GEDI and future waveform lidar missions. So, while we

do not assert that the WSCI is the final word, it does provide a means today that links space to ground-based measures of complexity from accelerating community efforts, such as GEO-TREES.

Minor points:

1. The researchers employ GEDI RHs as the fundamental predictor. While RH has greater fidelity to the original waveform, it suffers from light extinction / signal saturation. Perhaps I missed this, but were these factors accounted for? Were RHs otherwise normalized or transformed? How were negatives dealt with? Decision trees do not require normalization of inputs and the XGBoost models are not hindered by negative values, so no transformations were applied to the RH metrics. Signal saturation is embedded in the quality filters applied in our study (Table 2), particularly by applying high sensitivity thresholds to guarantee canopy penetration, as described in line 528.

2. Bullseye co-location still has associated sigma. How was this accounted for when co-locating GEDI footprints with ALS?

Since the collocation uncertainty is random, its average uncertainty is near zero in large samples. We did not apply any specific treatment for this and just followed the recommended procedures from the GEDI simulator, cited in the text (ref 51).

3. At several points (eg 126-129; 263; 204-211), the researchers make mention of variable importance via SHAP values, but did not provide these results. Instead, I'm squinting at tiny distributions on the figure margins (eg Fig. 1). It would be nice to see an analytical paper trail of sorts. This could be a good Supp table/figure and would support findings on how low-mid-upper canopy layers contribute to different complexity profiles in tropical versus temperate forests.

Thank you for suggesting this improvement. Figures 1 and 2 were changed, with feature importance added as individual panes. We also updated Figure 4 after a slight change in methodology, which facilitates the geographical interpretation of the feature importance profiles in Figure 2, splitting those results into 3 layers to aid their visualization in an RGB map. We further added Supplementary Figure 4 (line 265) showing examples of feature importance profiles from forests sampled from different regions and biomes, enabling a more detailed interpretation on how WSCI is predicted in different forest types.

4. I'm not sure how much the "lost" explained variance in the PCA models is due to WSCI (See my comments corresponding to row 247). I see alternative explanations for the same phenomenon, and think there are other ways, less susceptible to error propagation, to construct a null model to test that hypothesis.

Thank you very much for your suggestions on this. We updated the analysis following the comments on line 247. We also described the rationale behind that analysis while referring to potential effects of non-linear interactions unaccounted for in the PCA.

- Line 305: The question arises as to whether WSCI, as an index, captures any variation in complexity that is not already captured with existing standard GEDI metrics (RH98, AGBD, FHD, cover and PAI). The advantage of using a single index is well established as it facilitates the incorporation of structural complexity into ecological modeling and applications⁴. In addition, because WSCI uses the entirety of the waveform and infers horizontal variability, it is of interest to establish if it derives aspects of structure not captured by these other metrics (Supplementary Fig. 6). To assess this, we used the 3 first PCA components (which accounted for 98.8% of the total variance in the dataset) to perform a principal components regression (PCR) to predict WSCI. The PCR explained 96% of its variance with the unexplained 4% likely due to nonlinear interactions among the variables not accounted for in principal components space. We then

removed WSCI from our principal components analysis and found that the resulting first 3 principal components explained more than 99.5% of the variation in the data set, with a subsequent PCR of these 3 components explaining 85% of the variance in WSCI. Thus, based on these PCR analyses, the WSCI captures 11% of the variation not included in these other metrics. That said, it is possible that new models trained with these metrics and that account for nonlinear relationships could predict CE_{xyz} nearly as well. However, some of these metrics, such as AGBD³¹ and PAI³² rely on empirical calibrations and/or assumptions about canopy and ground reflectivity in their derivations, respectively. Using RH metrics directly avoids such issues and intrinsically provides a means to assess the contribution of elements of canopy structure as captured by the direct, cumulative energy returns at various heights to complexity.

5. The authors repeatedly refer to this as fine resolution. Well, maybe GEDI footprints count (though certainly not in comparison to ALS or TLS), but the subsequent analyses are clearly not. Nor are they even landscape scale. There are clear strengths to this analysis, but fine resolution is not one of them. Please try to clarify, and in the process, I would suggest addressing an obvious strength of the analysis: consistency over massive extents, two things TLS and ALS decidedly lack.

Thank you for helping us clarify this, as “fine” and “coarse” can indeed be ambiguous terms. Our use of fine resolution in the text was often in comparison to SSCIpot in the Results section. The revised manuscript has removed the comparison with SSCIpot from Results. We reviewed the remaining fine resolution mentions and either removed or explicitly linked them to the scale they refer to, such as GEDI footprints (e.g. lines 387, 444). We also highlighted consistency over massive extents explicitly in the text (line 139) as suggested.

Line-by-line comments:

25 - 3D "forest" or "canopy" complexity
Fixed.

26 - I read the results to say not that tropical forests are the "most" complex (which I interpret as having the greatest levels of complexity as measured by WSCI) but instead were, on net, "more" complex. This is a small but important point. While the median tropical forest is more complex than the median temperate one, temperate forests are still capable of the highest levels of complexity on par with those found in the tropics. For example, how do those Bornean Diptocarps compare with Redwoods? I imagine they are commensurate, and as WSCI and FHD are highly dependent on canopy height, I wouldn't be surprised if Redwoods had the world's highest levels of complexity ("most complex"), as they rank in biomass.
Changes made throughout the text (e.g. lines 26 and 133) to address this.

31 - I wouldn't call these spatial distributions "nuanced". First, I am not clear what a nuanced distribution is. Do you mean they feature finer resolution, greater precision, or more fine-grained patterns in spatial turnover? Elucidating on any of these characteristics didn't seem to me to be a goal of this paper. Instead, I think the "nuance" (not my word) comes from GEDI characterizations of vertical canopy profiles, which in this case likewise has a modeled horizontal component.
We agree, that was a poor choice of adjective and deleted “nuanced” from the sentence.

42 - Neither of those studies dealt with ecosystem function explicitly. Perhaps "biodiversity" is the word. Further, among the two, I would suggest substituting the EA paper for one recently published by the same lead in Environmental Research Ecology, which unlike those two, is an explicit test of GEDI for characterizing patterns in biodiversity (also sees rows 263 and 370).

Thank you for these suggestions. We have updated the references.

63 - Also, TLS signal saturation for larger/taller canopies.
Added.

74-78 - See major point #1. Here and below.
Please see our reply to major comment 1. Limitations have been reinforced and this material has been dropped from the Results.

88 - "empirical"
Fixed.

104 - Earlier you established the state of the field from TLS and ALS studies, so I wonder if "high-resolution" is the best way to distinguish your advancements in processing GEDI waveforms.
We have changed this.

105 - Did this study assess how "high-resolution estimates" of complexity "result from gradients in environmental drivers, seral stage and disturbance conditions." If not, I would either couch this sentence in terms of potential implications of the results and/or cite another study that does touch on these topics.
Thank you for the suggestion and we have changed the text accordingly.

124 - Linking to Supp. 1; surprising to see bias of 0.00 (versus -0.00, lol). In fact eyeballing the charts seems to definitely show an, acceptable IMO, bias. Can you confirm that these bias estimates are correct?
The bias estimates are correct. Skewed distributions of model residuals may cause an impression of bias. The negative sign was fixed in Supplementary Fig. 1.

126-129 - Where are these results? Fig 1 insets are too small to assess these relationships (for a reader). This could perhaps be a good Supp figure.
Figures changed with SHAP feature importance plots in individual panes.

115 - While you go into more detail in the Methods, I think a passing mention to the fact that GEDI shots are co-located with ALS would make your process more clear to the reader at this point.
Thank you. We clarified this in the revision (line 151).

147 - Why 6.3? Is this an artifact of the CExyz equations? I think this text, and any subsequent explanations, would be better located in the body text, not the caption.
Yes, that needed a clarification. It's a function of the horizontal diameter (25 m) which limits the maximum value that can be reached. An explanation was added to the main text (line 163) and removed from caption.

153 - Linking to Supp. 2; why is the Amazon Basin a continuous polygon and the other locations, more or less, discrete points?
We aggregated collocated samples into fixed size hexagons, as mentioned in the caption, to calculate local rmse and bias estimates. In the Amazon, the ALS point clouds were of smaller size and spread out across the entire region, while in other areas (e.g. NEON sites in the US) the point clouds covered larger, continuous areas, but limited to a single or few sites throughout the region.

180 - High quality (eg filtered) shots, no? Also, small typo: "shots and averaged"

Fixed.

204-211 - Here again, it would be nice to see an analytical paper trail of some sort (see minor points #3). Otherwise, I'm squinting at tiny distributions on the figure margins.

Figures changed for better visual interpretation.

225 - "canopy structure"

Fixed.

229 - How were these samples? Random? Stratified?

Thank you for pointing this out. The sampling was random and now noted in the text (line 289).

232 - Be explicit with PCA1 interpretation like you are with PCA2. Perhaps start off by saying more explicitly that "density" is driving PCA1 (if that's your interpretation).

We changed the text to address this.

236 - And/or height? FHD is correlated with height, to a large degree because it is highly determined by it. Half of Shannon's H is "S", which in this case is the total number of height bins, from which relative proportions are summed.

We now mention canopy height explicitly.

244 - Not a question, so no "?" needed.

Thank you.

247 - I'm not sure how much of this "lost" explained variance is due to WSCI, but rather the imperfect nature of linear models to capture variance in the data. Put another way, while PCA1-3 explains 99.5% (seems high, but okay I won't quibble) how much would it explain if you inverted it (PCA still retaining the WSCI information) to predict WSCI? I imagine somewhere in between 85-99.5% and likely towards the former. In other words, could you come to the same conclusion with WSCI retained.

We address this in our answer to minor point 4.

250 - What assumptions are you referring to? If light extinction / signal saturation is one, it could be a good assumption to retain.

The GEDI L2B ATBD describes the assumptions necessary to calculate the canopy gap fraction of GEDI footprints, used as input when deriving PAI. One of the assumptions relates to the canopy/ground reflectance, which cannot be measured by GEDI data alone, being derived by external datasets and assumed to be constant across large areas to allow for practical implementation. We now mention that in line 319:

- However, some of these metrics, such as AGBD³¹ and PAI³² rely on empirical calibrations and/or assumptions about canopy and ground reflectivity in their derivations, respectively.

254-260 - Capitalize acronyms

Fixed.

290 - Extra comma

Fixed.

297-308 - I would suggest dropping this section. Its underlying assumptions are not robust (comparing your gridded maps to an unvalidated highly speculative and likely problematic secondary map based on

limited ground samples). Further, the results aren't exactly compelling: similar patterns, with some fine scale deviations. I do not finish this section feeling like I learned something about the world like I did in other sections.

This section was dropped as suggested. Please see our answer to major point 1.

306 - Extra comma

Deleted.

307 - Missing an "as" before "well as"

Deleted. We apologize for these glitches that you have caught. Thank you.

308 - But GEDI is not swath mapping and has huge gaps in the tropics, so is this really fine-grained and local? No. But there are other advantages to GEDI ...

Yes, we agree, and deleted this.

326 - How much of this effect is driven by biome versus forest type (eg conifer and broadleaf)?

We rewrote the sentence to clarify this point:

- Line 407: In particular, the rate of increase of complexity with respect to height was variable by biome, largely driven by differences between PFTs captured by the different models. However, even within biomes, differences in these scaling patterns at finer scales were observed as the same model can vary its weights (Fig. 2) to predict complexity based on the characteristics of the input canopy profile.

329 - "6,"

Fixed.

335 - You hit resource abundance pretty hard, but might mention two other important axes hypothesized to drive community dynamics a la Grime: stress, and most importantly, competition.

Thank you for suggesting this. We have added the following:

- Line 419: In contrast, other forest biomes with more limited resource availability enable fewer dominant tree species to thrive optimally, which are often under stress induced by competition with individuals following similar growth strategies seeking the same resources³⁶.

345 - Does it though? There seem to be many studies that almost universally find consistent positive relationships. Also, there's an extra "." here.

Thank you. We have rephrased for clarity:

- Line 428: However, although generally positive associations between tree species diversity and structural complexity have been documented^{3,5,8,40}, the strength of these relationships remains uncertain. Therefore, the availability of high resolution WSCI estimates may facilitate further exploration to better understand how biodiversity and structure interact.

348-352 - Yes! Is this actually just a comparison to coarse environmental layers? And if so, why not just do that explicitly, and using better environmental data while you're at it! (I'm not suggesting you do that in this manuscript, btw)

Thank you for this good point. We dropped that analysis from the main results as suggested and further reinforced the generalization of climatic variables in the SSCI extrapolations in the introduction (line 93).

We also plan to investigate the environmental drivers of WSCI more deeply in the near future.

363 - Very glad you note this explicitly here. And, IMHO, it could be a more prominent point in earlier section (even abstract) without distracting from the strength of your results.

Thanks! We further reinforced the relationship between vertical and horizontal complexity throughout the text.

376 - Sparse tree cover, yes. But also sparse GEDI shot distributions.

Maybe relative to high latitudes where GEDI data is denser, but there is no difference in coverage between GSW and other regions. In fact, a lower proportion of high-quality footprints happen in moist tropical forests due to atmospheric interference caused by clouds than in GSW.

460 - "sensor agnostic index" is important enough of a point that you may consider mentioning it in the body text.

Mentioned earlier in the text (line 104).

470 - Redundant from paragraph before

Fixed.

503 - Were RHs normalized or transformed? Was anything done to address negative values?

Please see our reply to minor comment 1.

556- PCR? Did you mean "PCA"?

PCR = Principal Components Regression. The acronym was introduced at line 311.

Reviewer #2 (Remarks to the Author):

The authors present a newly developed forest structural complexity index (WSCl) based on GEDI spaceborne LiDAR data. This index is conceptually based on a recently published structural complexity index derived from airborne laser scanning data (CExyz by Liu et al. 2022) and applied to (nearly) globally available GEDI data to produce a global map of forest structural complexity. This dataset is then used to (1) compare WSCl with other GEDI derived structural metrics, (2) test scaling relationships between canopy height and structural complexity, (3) to compare the near-global forest structural complexity map with a previously published map by Ehbrecht et al. 2021, and (4) to compare the structural complexity of forests of different plant functional types (PFT) and how these are driven by structural differences in their under-, mid-, and overstory.

The manuscript is generally very well-written and addresses a timely and relevant topic. My main concerns are related to the methodological approach, some conclusions drawn from the results and whether the newly developed index WSCl provides additional information on top of what other GEDI derived structural indices provide already. Furthermore, I think that the manuscript requires more clarity regarding the terms used and how the authors define structural complexity. I will try to outline my concerns in more detail in the following and hope to provide some ideas on how the manuscript could be improved.

Thank you for your appreciation of our efforts and your careful review.

- The overall motivation and rationale of the study is clear. However, the introduction lacks a clear definition of what the authors are actually aiming to measure with WSCl. In line 356-357, the authors state that “there is no established definition of structural complexity so that the concept of a “true” complexity is fraught”. While I generally agree that a commonly accepted definition of structural complexity is lacking, there have been some attempts to define structural complexity that I think should be acknowledged here. In their review on relationships between plant diversity and structural complexity, Coverdale & Davies (2023, <https://doi.org/10.1111/1365-2745.14068>) discuss current definitions of structural complexity (see chapter 2 of their paper). The reviewed definitions by Ishii et al. (2004), Ehbrecht et al. (2021), Gough et al. (2019) and Atkins et al. (2018) all tend to point into similar directions, i.e. defining structural complexity as variability/heterogeneity of biomass/canopy element distribution in 3D space. Moreover, Ehbrecht et al. (2021) provided a theoretical framework on the drivers of structural complexity (Fig. 1) and LaRue et al. (2023) presented a theoretical framework for the ecological role of structural complexity. I think the introduction would strongly benefit from taking these definitions and state of research in this field into account.

Thank you for these helpful suggestions and references. We updated the text in the introduction to include the aforementioned state of the art research, particularly the first paragraph (line 36). To improve the introduction`s narrative and more clearly explain the motivation behind the development of the WSCl we added the following paragraph:

- Line 108: Lidar remote sensing of canopies can produce vast amounts of data leading to many different descriptors of canopy structure. This has led to the creation of indices of complexity that provide compact yet meaningful summaries of structural variability that may be readily mapped and interpreted by ecologists^{2,20-23}. Additionally, the use of a single index facilitates the incorporation of structural complexity into ecological modeling and applications⁴. While GEDI produces less data over its 25 m footprint than a TLS or ALS survey, its waveforms are nonetheless described by 100 RH metrics at each footprint. Hence, our motivation to provide an index of complexity from GEDI waveforms. Furthermore, GEDI only records vertical variations in

canopy structure, as given by variations in the amplitude of its returned waveform at any given height. There is thus considerable interest to assess the degree to which 3D complexity may be inferred from vertical variations in the waveform alone, under a hypothesis that vertical and horizontal complexity must be related.

- The study's main objective is the development of a footprint-scale Waveform Structural Complexity Index. What I think is lacking here are research questions or hypotheses. To improve the manuscript, I'd suggest to formulate research questions or hypotheses that are then being addressed in the results and discussion section.

Our work is largely data driven and exploratory/descriptive. There are strong elements of discovery in our research, nonetheless, as evidenced by our results, and while these could be framed, post-hoc, as hypotheses, we feel it made more sense to not do so. Instead, we follow the style of papers of similar scope, e.g. Lang et al. (2023): <https://www.nature.com/articles/s41559-023-02206-6> which describe the process of creating height maps from GEDI. We fully expect that subsequent efforts will indeed be framed within a context of a priori research questions enabled by the WSCI and our results.

- I do not fully understand how horizontal complexity can be derived from GEDI footprint data, when GEDI does not measure horizontal variability on footprint level, as the authors mention in line 120-121? Furthermore, I appreciate if the authors could present in more detail the difference between horizontal and vertical complexity and how both contribute to overall 3D complexity? I think it becomes surely clearer when looking at Liu et al.'s paper where the CExyz index is being introduced. However, to make the presented study more intuitively understandable, it should be clarified here without referring to Liu et al.'s paper.

Thank you for this suggestion. We have modified the paper to emphasize that WSCI is only able to infer horizontal complexity due to its strong relationship to vertical complexity, as supported by our results. For example:

- Line 160: Our results suggest that horizontal and vertical canopy structure are linked at the footprint scale, confirmed by the strong relationship between horizontal and vertical complexity observed in our ALS samples (Supplementary Fig. 1), which enable some variation in horizontal complexity to be inferred from vertical structure co-variates.

To make the relationship between vertical and horizontal complexity more intuitively understandable, we updated the introduction to address the relationship of horizontal and vertical structural variability to 3D complexity (lines 61, 70), the limitations in our understanding of that relationship (lines 85, 117) and also relate them in an ecological context:

- Line 175: These results further imply a functional dependency between vertical and horizontal structural attributes in the forest's overall complexity development. As canopy height increases, vertical layering also develops, such as through the recruitment of new individuals into the understory, diversifying the tree composition, increasing tree density^{3,27}, and consequently affecting the forest's horizontal and vertical complexity.

We also address how our inference of horizontal complexity is limited explicitly in the discussion and present research paths by which it may be improved:

- Line 466: One limitation of the WSCI is that it is only able to infer horizontal complexity within GEDI footprints through its association with explicitly measured vertical variability within footprints (i.e. the RH metrics). Our results showed that such an association exists, and that horizontal and vertical canopy variabilities are linked. For now, such inferential approaches may be the best that can be hoped for until lidar data with sufficient horizontal resolution (sub-meter) are available over

large portions of the Earth. The potential for using high-resolution stereo imagery as a substitute for lidar to resolve canopy features is increasing⁴⁴ though its efficacy in dense forests or to measure different vertical portions of the canopy is yet to be determined.

- The accumulated feature importance was divided into under- (\leq RH40), mid- ($>$ RH40 and \leq RH90), and overstory ($>$ RH90) layers. The selection of the respective RH values as breakpoints for the different layers strongly affects the results. What was the basis for the selection? Was the classification based on an approach presented in another study? Other studies, e.g. Willim et al. 2020 (<https://doi.org/10.3390/rs12121907>) used other classifications (e.g. RH33, RH66 and $>$ RH66). Using different breakpoints for classification would likely yield different results and thus probably result in drawing different conclusions.

Thank you for this suggestion. We changed the thresholds to RH33 and RH66 to maintain an equal size of the different layers, which did not significantly affect the results. We also changed the text to avoid using the terms understory, midstory and overstory, as those terms are often used in an ecological context where their meaning is variable. We now simply call these lower, middle and upper waveform layers to clarify where the RH metrics used in the models originate from. We hope this makes interpretation of the analyses more straightforward. Ultimately, the RH thresholds used in Figure 4 are to facilitate visualization of the feature importance profiles in Figure 2 and their variation on a global map. Following reviewer`s 1 suggestion, we also added Supplementary Figure 4 to the revision to enable a more direct comparison of those profiles in forests sampled from different regions and biomes.

Also, please recall that GEDI RH metrics are not a fraction of total canopy height (line 159), but rather a fraction of total returned energy. As such, the same RH metric can correspond to different canopy height percentiles in different forests. The study by Willim et al. (2020) applied reasonable, but also somewhat arbitrary thresholds based on total height for the different layers. This is understandable for a study comparing similar forest types, but this approach may not be appropriate where forests of widely varying canopy heights and covers are included.

-The authors show that WSCI correlates well with other structural metrics, especially RH98 and Foliage Height Diversity, and particularly ask the question whether WSCI “captures any variation in complexity that is not already captured with existing standard GEDI metrics”. If 3 principal components from the PCA explain 85% of the variance in WSCI, can we confidently conclude that WSCI “captures variation not included in these other metrics”? My question is, would we come up with different conclusions about the global patterns of structural complexity if we simply used FHD? Would WSCI allow us to identify differences between forest types that we wouldn’t see by using FHD?

We extended the PCA analysis following suggestion from reviewer 1 (line 305), updated Supplementary Figure 3 to enable comparisons with FHD and added Supplementary Figure 6 (line 303) to demonstrate how WSCI is a more effective integrator of structural information than those other variables. Supplementary Figure 6 shows how WSCI better captures variation in cover and PAI than FHD does, while maintaining strong relationship with height and vertical features like FHD itself. Also, FHD does not measure horizontal complexity within a footprint, but this is inferred in the modeling of WSCI.

-A few thoughts about index construction (figure 5a): PC1 (with Cover and PAI) quantifies something like a density component, while PC2 (with WSCI, FHD and RH98) quantifies something like a vertical structure component. Other indices, like e.g. SSCI, are also composed of a kind of density component (perimeter-area-ratio of polygons) and vertical structure (ENL, which is basically the same as FHD, but with a different scaling ($\exp(\text{FHD})$). See also line 38-39. Wouldn’t WSCI rather provide added information and be more complementary to other existing indices (i.e. more different from Rh98/FHD), if it was based on a combination of Cover/PAI and FHD/RH98. In the PCA, it would then probably load on the axis

between Cover/PAI and FHD/RH98. I am particularly wondering if the authors have tried or experimented with different approaches for index construction. If so, it would be great if these could also be presented in the SI.

Thank you for this interesting discussion. We have not attempted to construct indices from other approaches at this time, though future work, either by ourselves or others, will likely do so. We do reinforce in the text the reliance on a horizontal complexity component to estimate 3D structural complexity, which GEDI cannot measure directly but may be able to infer, as a unique element. We added a paragraph in the discussion to explain the reasoning behind our model-based approach:

- Line 393: Structural complexity is inherently linked to 3D space^{1,2,4}, hence our choice of a metric that separates complexity into vertical and horizontal components²⁰. Metrics such as FHD simply ignore the horizontal component. GEDI waveforms only provide direct measurement of the vertical component, leaving the horizontal component to be inferred indirectly through our model-based approach. This approach is analogous to the modeling of above ground biomass (AGB), which relies on in situ plot measurements to train prediction models based on measurements (e.g. height) that are indirectly related to AGB³¹. High-resolution ALS data are widely available and provide sufficiently detailed measurements to quantify structural complexity accurately and precisely, and therefore represent the best source of training data. Advanced machine learning models can also integrate variables from high-dimensional space to capture patterns beyond traditional ecological metrics like FHD or PAI. Our model-based approach has additionally provided insights into the sensitivity of GEDI waveforms to 3D structural complexity and uncertainty estimates for use in ecological inference, highlighting areas to target for improvement (e.g., tropical savannas).

-Comparing the actual structural complexity as measured by WSCI to the potential structural complexity measured by globally modelled SSCI data is very interesting. Despite a low R^2 , I find it quite surprising that WSCI and SSCIpot show a significant correlation, given the fact that (1) the two indices quantify structural complexity from completely different perspectives based on completely different approaches, and (2) because SSCIpot quantifies the level of structural complexity that is possible in the absence of disturbances. Against the background that the largest part of the world's forests show signs of anthropogenic disturbance, wouldn't it make sense to (additionally) focus on a comparison of WSCI and SSCIpot for old-growth or primary forests? Generally, I am concerned about drawing conclusions from a , higher integrity, global comparison -as the authors themselves also point out in line 354- given the fact , that both indices are very different by design. The comparison could be more straightforward, if WSCI had a design similar to SSCI (see also my earlier comment on index construction). Thus, additionally focusing a comparison on primary forests could potentially eliminate a few confounding factors. When discussing the results, I also suggest to put more emphasize on the potential of comparing actual and potential complexity to quantify forest structural integrity or degradation, and outline which further work is needed to facilitate proper comparisons.

Thank you for these comments. We quite agree that further work is needed to facilitate proper comparisons. We followed your suggestion for improving the comparison by limiting it to intact forests that might be expected to be in more of a potential state. However, due to concerns also raised by reviewer 1 we decided to drop this section from Results in the paper. Instead, we provide an analysis of the two for intact forests in the Supplement. Maps of intact forests are more readily available than, say, maps of mature/old-growth forests. We do address the comparison between WSCI and SSCIpot directly in the Discussion in the main body of the paper as follows:

- Line 433: The WSCI is derived from structural attributes and is influenced by disturbances underlying GEDI observations. As an effective integrator of structural information (Supplementary Figure 6), the WSCI may be useful to assess forest degradation and structural integrity (the ecosystem's capacity to maintain its structure, function and composition relative to its natural

range of variation⁴¹). The $SSCI_{pot}$ product estimates potential structural complexity¹⁶, and its comparison with actual complexity (WSCI) may be informative; for example large differences between $SSCI_{pot}$ and WSCI where the current complexity of the forest is much less than potential could imply forest degradation and a loss of integrity. However, caution must be exercised when comparing indices that, while designed for the same purpose, may not be equivalent and thus yield unreliable interpretations. In our case here, we found the two indices were only weakly correlated, even when limiting comparisons to intact forests⁴² (Supplementary Fig. 9). This lack of agreement is likely indicative of the divergent means by which the indices were derived. The WSCI uses structural attributes directly to model complexity at fine scales (GEDI footprints) spatially across the domain of GEDI observations. In contrast, the $SSCI_{pot}$ relies on climate variables at much coarser scales. Additionally, the $SSCI_{pot}$ models are strongly influenced by boreal forests, which are not observed by GEDI. Given the strong concentration of intact forests in boreal regions it is perhaps not surprising the two indices exhibit weak correlation (Supplementary Fig. 9). Future research should attempt to derive an equivalent WSCI index from the ICESat2⁴³ mission, which has excellent coverage in boreal regions, which if accurate, could provide the means for assessing deviations from potential as given by $SSCI_{pot}$.

I enjoyed reading and reviewing the manuscript. I hope my comments may help to improve the paper. Thank you for your careful attention to our paper. Your comments have been invaluable and are greatly appreciated.

Reviewer #3 (Remarks to the Author):

Dear Authors,

I really enjoyed reading your manuscript and think you conducted a state-of-the-art study to analyze the structural complexity of forests globally. I think it is a valuable demonstration on how to turn technical data into useful environmental information and their interpretation.

Thank you for your encouraging words.

Here are some (minor) thoughts and suggestions about the manuscript that I think are worth including:

1. Readers that are unfamiliar with the GEDI system might wonder, why all the maps show no data around the poles and why you did not include boreal forests in your analysis. I suggest adding the technical information about GEDI data availability in the introduction and briefly discuss the implications of excluding boreal forests in your research.

We make the GEDI latitudinal range explicit in the introduction (line 99) and addressed the importance of boreal forests in the discussion (lines 446, 489) and how the use of other remote sensing data may derive this in the future.

2. Usually Light detection and ranging is abbreviated as LiDAR, I personally don't mind the all lower case lidar but some readers might.

Yes, the continuing variations of this in papers is a bit maddening. We opted to keep lowercase use of "lidar", similar to "radar", as it is a well-established technology among the remote sensing community and hope not many are irritated by it.

3. From a modelling point of view, one thing you should add and discuss in the manuscript is the impact of the distribution of the ALS training data. Why is your model performance not as good in regions outside the tropics (as seen in Figure 3b and Supp. Figure 2)? Why is it especially low in the western USA - despite the fact that this is a region the model should be quite good because there are ALS data available? And what implications does this error distribution have for regions with no ALS data available (e.g. Europe or Central Asia)? In other words, is your model trustful in these areas? I think for a better understanding and writing of this point, it could be worth it to move Supp. Figure 2 into the main manuscript or show a small ALS distribution map in Figure 3.

In the north-western US, where temperate forests dominate, our models exhibit strong performance. However, as we move towards the south-west, the landscape shifts towards desert terrain, leading to a savannah-like tree cover (GSW) and subsequently lower model performance.

Our models, trained by PFT, are designed to generalize across regions. For instance, the evergreen broadleaves model draws samples from both tropical and temperate biomes, enabling it to extrapolate beyond its training domains. We've ensured robustness through rigorous spatial cross-validation criteria during model training, as detailed in the methods (lines 623-631) and highlighted in our results (line 151) sections. Thus, metrics like R², rmse, and bias reflect the models' performance on unseen data during training, bolstered by our extensive training dataset, instills confidence in their geographical generalization.

As mentioned earlier our training data originates from the GEDI mission and is one of the best such database compiled: the Forest Structure and Biomass Database (FSDB). The FSDB is used for

calibration and validation of GEDI observations as well as development of the GEDI biomass models (Duncanson et al. 2022), which employed a similar approach to segment the world into prediction strata, training models within comparable regions limited by data availability. Future updates to the FSDB are ongoing and are targeting potential geographic gaps, such as in Asia. This expansion could enhance the WSCI models by incorporating training data from previously unobserved regions and may facilitate the development of specialized models tailored to finer domains than those covered by PFT.

To enhance clarity, we've included an inset in Figure 3, pinpointing the locations of ALS training sites.

4. Two random spelling errors I encountered during reading: l88 - empirically -> empirical; l116 - database -> data

Fixed.

REVIEWERS' COMMENTS

Reviewer #1 (Remarks to the Author):

I am happy with this revision, especially the fair treatment of SSC|pot and the elaboration on the ALS-derived nature of WSCI. I liked the AGBD analogy, which I think drives home the point that modeling compromises are necessary where underlying data lack the requisite information (e.g. horo complexity in a single GEDI waveform). I likewise appreciate the discussion of RH metrics as agnostic predictors.

A few (very) minor suggestions:

24: assessment of what?

30: I think a critical (missed) point for the abstract is that this analysis is the basis for a WSCI data product, of great interest to the research and conservation communities

35: vary,

66: "in" > "on"

68-69: to this I would add, fragmentation among multiple datasets (not always the case, but often is)

124: great point

133-136: i get the point, but the wording could be improved

147: "canopy top" > "upper strata"

160: great improvement - I can finally see the FIs!

245: canopies'

245: "present" > "facilitate"

276: clarify "its"

332: and are highly dependent on RH100 (e.g. n height bins)

396: "inter-footprint"?

398: "structural complexity"

399: did you mean "larger"?

417-420: can you cite the dataset?

Reviewer #2 (Remarks to the Author):

I appreciate how the authors have revised the manuscript and think the manuscript has improved a lot. However, I still have some questions and think there is a need for further clarifications and corrections. Please see a few line comments in the following:

L54 – I suggest to replace “structurally complex” with “forest structure”. A forest can also be low in structural complexity, still structure determines microclimatic conditions or niche space.

L93 – We have cross-validated our results by excluding up to 30% of the data. Still, the model was quite robust without substantial reductions in the explanatory power of the model. Please note that

it was 20 sites, not 19.

L120 – I appreciate that the authors now address the topic of structural complexity definitions already in the first sentences of the introduction. However, as a reader I am still a bit puzzled to understand what horizontal and vertical complexity mean here. CExyz is an entropy-based metric. It would be great if the authors could mention that here. Any brief explanation what horizontal and what vertical complexity is and how they jointly characterize 3D complexity would be appreciated.

L122 – The authors mention 800,000 values of CExyz from ALS, whereas Table 2 shows that the ALS data comes from 11 sites. To put the 800,000 values into context, it would be great to clarify, e.g. in SI Fig. 2 or in the methods section, how the 800,000 measurements are distributed across the 11 sites. Looking at SI Fig. 2, do I understand correctly that the largest share comes from ALS data from the Amazon?

L204 to 214 – I would say the information given here is rather discussion and could be moved to the discussion section or be placed in the SI.

L269 – I find it difficult to interpret the triangle without axis descriptions and units. Similar approaches are used e.g. for soil texture classifications, where each of the three axes goes from 0 – 100%. I think it could make sense to add that.

L290 to 297 – Interesting! In the first round of reviews we discussed about complexity being a function of both density and distribution of foliage and woody components. I agree that PC1 aligns with density. For PC2, I suggest to avoid the term complexity here and rather speak of vertical stratification (and therewith foliage distribution) and canopy height. Then it becomes clearer that complexity is a function of density and distribution.

L 317 – 11 % of variation explained by WSCI that is not explained by other metrics is not a lot. I appreciate the authors now provide a figure and some more explanations in the text. SI Fig. 6 is also very helpful. There is surely some potential to develop WSCI further in the future, so that it explains more variation that cannot be explained by other structural metrics.

L445 – I think I don't understand why the SSCIpot models should be strongly influenced by boreal forests, when there were only two boreal forest sites included in the analysis to build the model?

I also think it was a good decision to exclude the SSCIpot analysis from the study. Even though I disagree with most points raised by reviewer 1 in that matter, I agree that the SSCIpot analysis was distracting. Still, I appreciate the additional analysis (with only intact forests) and the general discussion of the topic. I think there is definitely some potential for some further investigations in the future.

Thanks for the opportunity to review and comment on this manuscript.

Best regards

Martin Ehbrecht

Reviewer #3 (Remarks to the Author):

Thank you for addressing my suggestions. I have no further comments.

Reviewer #1 (Remarks to the Author):

I am happy with this revision, especially the fair treatment of SSC|pot and the elaboration on the ALS-derived nature of WSCI. I liked the AGBD analogy, which I think drives home the point that modeling compromises are necessary where underlying data lack the requisite information (e.g. horizontal complexity in a single GEDI waveform). I likewise appreciate the discussion of RH metrics as agnostic predictors.

Thank you very much for your thorough comments and suggestions and for reading through the manuscript again. We greatly appreciate your efforts on our paper.

A few (very) minor suggestions:

24: assessment of what?

Fixed (line 10): “[...] global assessment of actual forest structural complexity [...]”

30: I think a critical (missed) point for the abstract is that this analysis is the basis for a WSCI data product, of great interest to the research and conservation communities

Thank you for the suggestion. We now finish the abstract stating that (line 19): “Ultimately, the GEDI Waveform Structural Complexity Index data product derived from our analyses provides a valuable tool for researchers and conservationists by combining various aspects of canopy structure into a single, widely useful and easily interpretable metric.”

35: vary,

Fixed

66: "in" > "on"

Fixed

68-69: to this I would add, fragmentation among multiple datasets (not always the case, but often is)

Added (line 58): “[...] the limited spatial extent, lack of global availability and fragmentation among ALS and TLS datasets [...]”

124: great point

Thank you.

133-136: i get the point, but the wording could be improved

Fixed (line 129): “As part of our analyses, we also created separate models to predict horizontal and vertical complexity at the PFT level. This approach helps us determine if horizontal canopy structure, which GEDI waveforms do not measure, can be inferred based on its relationship with vertical structure.”

147: "canopy top" > "upper strata"

Fixed

160: great improvement - I can finally see the FIs!

Thank you.

245: canopies'

Fixed

245: "present" > "facilitate"

Changed to "encompass" (line 254): "[...] reflecting taller canopies' potential to encompass more 3D space in which elements such as canopy layering may develop [...]"

276: clarify "its"

Changed to "WSCI's"

332: and are highly dependent on RH100 (e.g. n height bins)

Fixed (line 349): "Metrics such as FHD simply ignore the horizontal component and are highly dependent on the number of height bins, which is determined by the canopy's top height."

396: "inter-footprint"?

Fixed

398: "structural complexity"

Fixed

399: did you mean "larger"?

Fixed

417-420: can you cite the dataset?

Cited

Reviewer #2 (Remarks to the Author):

I appreciate how the authors have revised the manuscript and think the manuscript has improved a lot. However, I still have some questions and think there is a need for further clarifications and corrections. Please see a few line comments in the following:

Thank you very much for your kind words and thorough review of both versions of our manuscript.

L54 – I suggest to replace “structurally complex” with “forest structure”. A forest can also be low in structural complexity, still structure determines microclimatic conditions or niche space.

Thank you. This is a good point and we have changed the phrase.

L93 – We have cross-validated our results by excluding up to 30% of the data. Still, the model was quite robust without substantial reductions in the explanatory power of the model. Please note that it was 20 sites, not 19.

Fixed (line 72): “[...] their models relied on 294 TLS point clouds distributed across 20 sites in different continents, producing a global map extrapolated from a small pool of training samples. Although their models underwent cross-validation, the limited number of samples restricted the extent of validation, which could contribute to overgeneralization of complexity as a function of climate variables.”

L120 – I appreciate that the authors now address the topic of structural complexity definitions already in the first sentences of the introduction. However, as a reader I am still a bit puzzled to understand what horizontal and vertical complexity mean here. CE_{xyz} is an entropy-based metric. It would be great if the authors could mention that here. Any brief explanation what horizontal and what vertical complexity is and how they jointly characterize 3D complexity would be appreciated.

Changed (line 95): “Here we develop a footprint-scale Waveform Structural Complexity Index (WSCFI). This index is established by modeling the empirical relationships between GEDI RH metrics and an existing metric of 3D structural complexity (CE_{xyz})²⁰ derived from ALS point clouds. CE_{xyz} is an entropy-based measure that captures both horizontal and vertical complexity. Horizontal complexity refers to the spatial distribution of canopy structures within a footprint, while vertical complexity describes the distribution of vegetation layers from the ground to the canopy top. Together, these components provide a comprehensive characterization of the 3D structural complexity of the forest canopy.”

L122 – The authors mention 800,000 values of CE_{xyz} from ALS, whereas Table 2 shows that the ALS data comes from 11 sites. To put the 800,000 values into context, it would be great to clarify, e.g. in SI Fig. 2 or in the methods section, how the 800,000 measurement are distributed across the 11 sites. Looking at SI Fig. 2, do I understand correctly that the largest share comes from ALS data from the Amazon?

Thank you for pointing this out. We added a column to Table 2 with the number of GEDI shots intersected by each ALS project. Recall that each ALS project does not correspond to a single site, but rather a network of ALS surveys within a geographical domain covering several widespread sites. The INPE project over the Amazon contains the largest number of sites which are sparsely

distributed, however the NEON project has much larger ALS coverage per site. The column added to Table 2 makes that information more transparent to the reader.

L204 to 214 – I would say the information given here is rather discussion and could be moved to the discussion section or be placed in the SI.

Transferred to discussion section (line 371).

L269 – I find it difficult to interpret the triangle without axis descriptions and units. Similar approaches are used e.g. for soil texture classifications, where each of the three axis goes from 0 – 100%. I think it could make sense to add that.

Figure 4 provides a visual interpretation on a geographical context of what is quantitatively displayed in Figure 2. The color triangle should help inform the reader on what different color combinations represent, thus we refrained from adding too much detail to the color scale to avoid distraction from the RGB map itself. Analytical interpretations of the different layer combinations can be better drawn from Figure 2 and SI Figure 4. Nevertheless, to improve readability we marked each subplot in the figure, addressed each one in the caption and added range + units of each band in the triangle's edges.

L290 to 297 – Interesting! In the first round of reviews we discussed about complexity being a function of both density and distribution of foliage and woody components. I agree that PC1 aligns with density. For PC2, I suggest to avoid the term complexity here and rather speak of vertical stratification (and therewith foliage distribution) and canopy height. Then it becomes clearer that complexity is a function of density and distribution.

Thank you for this suggestion. We have changed the text accordingly.

L 317 – 11 % of variation explained by WSCI that is not explained by other metrics is not a lot. I appreciate the authors now provide a figure and some more explanations in the text. SI Fig. 6 is also very helpful. There is surely some potential to develop WSCI further in the future, so that it explains more variation that cannot be explained by other structural metrics.

Thank you for this comment and we agree about future development regarding WSCI.

L445 – I think I don't understand why the SSCIpot models should be strongly influenced by boreal forests, when there were only two boreal forest sites included in the analysis to build the model?

Changed to (line 411): Additionally, the SSCIpot models are also influenced by boreal forests, which are not observed by GEDI but are estimated in the SSCIpot map.

The boreal samples would have a minimal impact on fitting the SSCIpot models, but these models make predictions across the entire boreal zone -a region we cannot directly compare to GEDI estimates. If GEDI predictions were available throughout the boreal zone, the relationship between the two maps would likely be different.

I also think it was a good decision to exclude the SSCIpot analysis from the study. Even though I disagree with most points raised by reviewer 1 in that matter, I agree that the SSCIpot analysis was distracting. Still, I appreciate the additional analysis (with only intact forests) and the general

discussion of the topic. I think there is definitely some potential for some further investigations in the future.

Thanks for the opportunity to review and comment on this manuscript.

Thank you again for your careful review and we indeed hope to explore these issues further in the future.

Best regards
Martin Ehbrecht

Reviewer #3 (Remarks to the Author):

Thank you for addressing my suggestions. I have no further comments.

Thank you for reviewing our manuscript!